# Batch alignment of single-cell transcriptomics data using deep metric learning

Xiaokang Yu[1,4], Xinyi Xu ®[2,4], Jingxiao Zhang ®[1] ✉ & Xiangjie Li ®[3] ✉

scRNA-seq has uncovered previously unappreciated levels of heterogeneity. With the increasing scale of scRNA-seq studies, the major challenge is correcting batch effect and accurately detecting the number of cell types, which is inevitable in human studies. The majority of scRNA-seq algorithms have been specifically designed to remove batch effect firstly and then conduct clustering, which may miss some rare cell types. Here we develop scDML, a **d**eep **m**etric **l**earning model to remove batch effect in **sc**RNA-seq data, guided by the initial clusters and the nearest neighbor information intra and inter batches. Comprehensive evaluations spanning different species and tissues demonstrated that scDML can remove batch effect, improve clustering performance, accurately recover true cell types and consistently outperform popular methods such as Seurat 3, scVI, Scanorama, BBKNN, Harmony et al. Most importantly, scDML preserves subtle cell types in raw data and enables discovery of new cell subtypes that are hard to extract by analyzing each batch individually. We also show that scDML is scalable to large datasets with lower peak memory usage, and we believe that scDML offers a valuable tool to study complex cellular heterogeneity.

Single-cell RNA sequencing (scRNA-seq) technology has been developed to characterize gene expression profiles at single-cell resolution, which improves the detection of known and novel cell types, as well as the understanding of cell-specific molecular processes and disease dysregulation within heterogeneous tissues. However, the widespread application of scRNA-seq has generated many large and complex datasets, which presents new computational challenge for integrating datasets from different batches and platforms[1–4].

A fundamental task in scRNA-seq data analysis is to cluster cells into different groups as candidate cell types or cell states. This task may be simple for the dataset from a single source, but is very difficult for the multi-source data due to the challenging characteristics of batch effect, especially for detecting some small clusters. Although several methods have been developed to remove batch effect in scRNA-seq analysis, most of them aim to remove batch effect in the embedding space but without considering the clustering structure or the local structure in the dataset. Popular methods such as Seurat[5–7]

and MNN[8] rely on the mutual nearest neighbor approach to remove the batch effect, but MNN can only analyze two batches at a time, so its performance is affected by the batch correction order, and it quickly becomes computationally infeasible when the number of batches increases. As such, the researchers introduced fastMNN[9], resulting in significant improvements in both computational speed and accuracy. Two other methods, Scanorama[10] and BBKNN[11], also search for MNNs in the dimensionally reduced space and use them in a similarity-weighted manner to guide batch integration. Moreover, two supervised MNN methods (SMNN[12], iSMNN[13]) were developed for batch effect correction of scRNA-seq, but these two methods require exactly the same cell type between different batches. Zou et al. presented DeepMNN[14] based on residual neural network that minimizes the batch loss, i.e., the sum of the Euclidean distance between MNN pairs in the PCA subspace. Based on the benchmark studies[15,16], due to its significantly shorter runtime, Harmony is recommended as the first method to try, with the other methods as viable alternatives.

[1]Center for Applied Statistics, School of Statistics, Renmin University of China, 100872 Beijing, China. [2]School of Statistics and Mathematics, Central University of Finance and Economics, 100081 Beijing, China. [3]Changping Laboratory, 102206 Beijing, China. [4]These authors contributed equally: Xiaokang Yu, Xinyi Xu. ✉e-mail: zhjxiao@ruc.edu.cn; xiangjieli@cpl.ac.cn

Benchmark study from Luecken et al.[17] also suggests using scANVI[18], scVI[19] and scanorama[10] on complex integration tasks, but the semi-supervised mode of scANVI and the time-consuming issue of scVI hinder the application. Although INSCT[20] is scaled to the large atlas and can conduct semi-supervised analysis that enables users to classify unlabeled cells by projecting them into a reference with annotated labels, it has poor robustness and reproducibility. The performance of BERMUDA[21] depends on MetaNeighbor[22], which limits its scalability and accuracy. Liger[23] aims to remove technical variation using integrative non-negative matrix factorization, but its procedure needs chosen reference dataset (typically the set with the largest number of cells). scVI[19] and CarDEC[24] are also designed for both removing batch effect and denoising gene expression simultaneously, but a recent study shows that the corrected counts output by the decoder layer of these two methods are usually over-denoised[25], which turn almost all zero expression values into non-zero.

Most existing methods first remove batch effects and then cluster cells. However, this procedure has the disadvantage that removing batch effect may lead to loss of the original rare cell type information. Therefore, in this article, we begin with the prior clustering information of the original data, and then take advantage of the nearest neighbor (NN) information intra and inter batches in the framework of deep metric learning with triplet loss, to properly recover true cell types and remove batch effects by learning a low-dimensional representation of data. Most importantly, scDML is not affected by the batch integration order. In the initial clustering, we first cluster cells at a high resolution to guarantee the initial clusters include all subtle and potential novel cell types, and then proposed a merging criterion to optimize the final number of clusters. This algorithm combines the advantages of graph-based clustering and hierarchical clustering methods, and simultaneously removes batch effect by pulling points with the same label close together while pushing away points with different labels.

In this work, we apply scDML to several simulated datasets and a wide range of real scRNA-seq datasets from different species and tissues to demonstrate its effectiveness. We also compare scDML with existing state-of-the-art integration methods. The results show that scDML can recover the biological hierarchy underlying the data, achieve high clustering accuracy under a fixed number of clusters, and also scale well to large datasets. Additionally, scDML is developed based on the framework PyTorch and preprocesses using the popular scRNA-seq analysis framework Scanpy[26], which is freely available via https://github.com/eleozzr/scDML.

## Results

### Overview of scDML and evaluation

scDML is designed to align multiple batches of single-cell transcriptomic data, which enables the discovery of rare cell types that might be hard to extract by analyzing each batch individually. The workflow of scDML is exhibited in Fig. 1. After preprocessing the scRNA-seq data (including normalization, log1p transformation, finding highly variables genes, scaling data, PCA embedding), we first used a graph-based clustering algorithm at a high resolution. Then, we used k-nearest neighbor (KNN) and mutual nearest neighbor (MNN) information within and between batches to evaluate the similarity between cell clusters and built a symmetric similarity matrix with a hierarchical structure. Cutting the tree at a given height (a given number of clusters) usually yields a partitioned clustering at a selected precision. scDML applies a merging rule, which is different from BERMUDA[21], and can generate a more stable result. Moreover, scDML utilizes the idea of hierarchical clustering to merge clusters one by one. For the detailed merging procedure please refer to Algorithm 1 in Supplementary Note. In this article, we used the number of true cell types as the cut-off for all datasets analyzed, to evaluate the performance of all compared methods. To successfully remove batch effects, we adopted deep triplet learning by considering the hard triplets, which helps to learn a low-dimensional embedding that properly accounts for the original high-dimensional gene expression and removes batch effects simultaneously (Algorithm 2 in Supplementary Note).

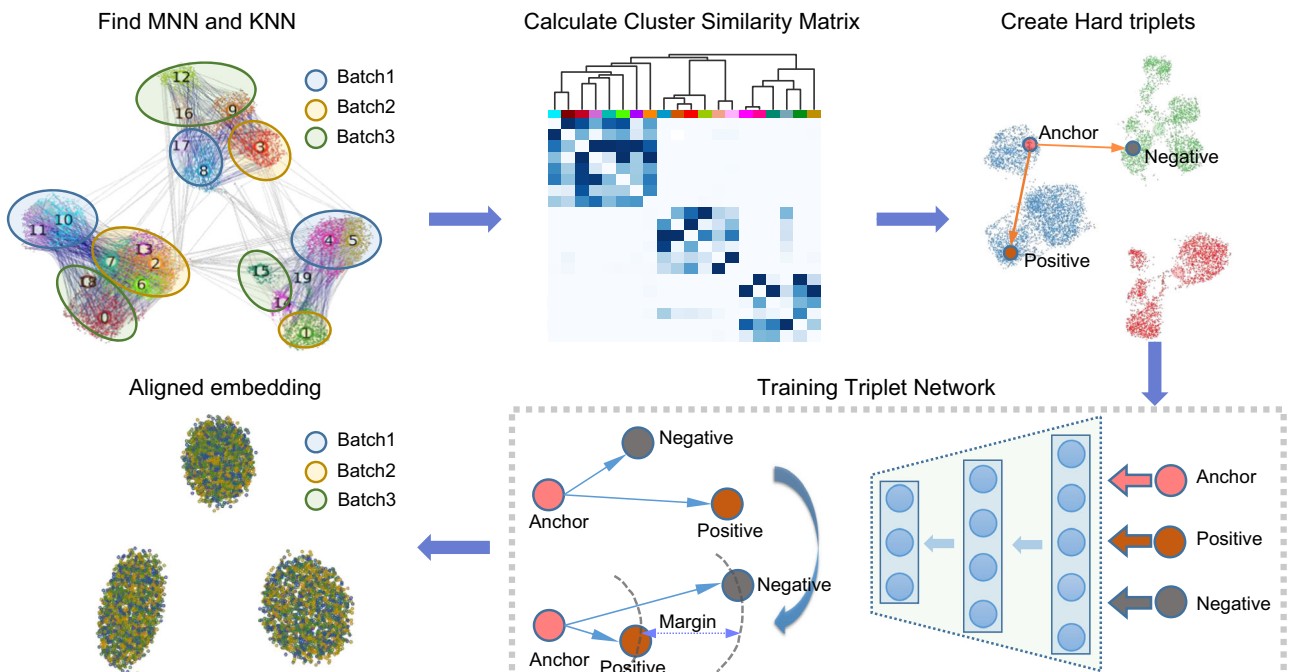

**Fig. 1 | Overview of scDML for merging clusters and removing batch effects in scRNA-seq data.** MNN means mutual nearest neighbors and KNN means k-nearest neighbors. The procedure of constructing cluster similarity matrix and the developed merge rule aim to preserve the original cluster information, including some subtle biological clusters. The goal of triplet network is to minimize the distance between the anchor-positive pair (from same cluster) while maximizing the distance between the anchor-negative pair (from different clusters).

The commonly used UMAP[27] method was utilized to visualize the results of scDML and all compared methods. In addition, three metrics (ARI[28], NMI[29], ASW_celltype[15]) were adopted to evaluate the clustering performance and three metrics (iLISI[30], BatchKL[31], ASW_batch[15]) were used to evaluate the ability to remove batch effect (for full names and definitions of the metrics, see Methods). To demonstrate the strength and scalability of scDML, we analyzed multiple scRNA-seq datasets from different species and tissues generated with different scRNA-seq protocols (Supplementary Data 1). The performance of scDML was compared with 10 methods aimed at batch effect correction including Seurat 3[7], Harmony[30], Liger[23], Scanorama[10], scVI[32], BERMUDA[21], fastMNN[9], BBKNN[11], INSCT[20], CarDEC[24] (Supplementary Table 1), as well as the raw data before batch integration. The parameters used for scDML are listed in Supplementary Tables 2, 3. Our results showed that scDML consistently performs better than these existing methods, especially for the ability to preserve rare cell types.

## scDML removes batch effect and preserves true structure in simulated data

To demonstrate the effectiveness of scDML, we applied our method and 10 state-of-the-art competitors to two simulations. In simulation 1, the data is generated by Luecken et al.[17], consisting of 4 cell types across 4 batches. The UMAP plots show that there is severe batch effect in the raw data, while only scDML thoroughly mixed cells from different batches and removed the batch effect (Fig. 2a). All other methods separated cells both by cell type and by batch, whereas scDML splits cells just by cell type (Fig. 2b) and preserved true cell types in the clustering result (Fig. S2a). Moreover, scDML resulted in the highest ASW_celltype (Fig. 2c), ARI (up to 1.0), and NMI (up to 1.0) (Fig. 2e), indicating its ability to improve clustering accuracy. Although scDML ranked third in the comprehensive evaluation of two batch mixing metrics iLISI and BatchKL (Fig. 2d), the top two methods Liger and INSCT failed to recover true cell types in their UMAP embeddings.

In simulation 2, the data is also generated by Luecken et al.[17], which has 7 cell types across 6 batches. All methods well mixed different batches except for BBKNN (Fig. S1a), but only scDML, CarDEC, Harmony, fastMNN, Scanorama, and scVI presented true cell types and clean clusters (Figs. S1b, S2b), with the highest ARI and NMI (Fig. S1e). scDML also led to the second highest ASW_cellytpe, only slightly inferior to INSCT (Fig. S1c), however, according to the UMAP plots, INSCT obviously corrupted the original data structure. It is demonstrated in the simulations that scDML outperforms other competing methods in batch effect removal, cell type preservation and clustering accuracy.

## scDML removes batch effect and preserves true structure in real datasets

We next evaluated the performance of scDML on several real datasets. The first is the mammary epithelial cell dataset from three independent studies[33–35], consisting of 3 batches and 3 cell types with 9288 cells in total. It is shown that scDML made cells separated by cell type and mixed by batch (Fig. S3a, b), provided clean clusters (Fig. S3c), and realized accurate clustering with top-ranking ARI, NMI, and the second highest ASW_celltype (Fig. S3e, f). Although Liger and BERMUDA displayed high batch mixing metrics (Fig. S3d), they wrongly split cell types *basal* and *luminal_progenitor* in their UMAP embeddings. Although CarDEC had the highest ASW_celltype, it had the worst batch mixing metrics BatchKL and iLISI, where its UMAP embeddings pulled together but failed to mix different batches. The merge order in each step of scDML suggests that true cell types were recovered when the number of clusters was set to 3 (Fig. S4).

To verify the flexibility of scDML in the extreme case where a cell type exists only in a single batch, we artificially removed cell type *basal* from two batches. scDML presented similar desirable performance as in the real mammary epithelial dataset (Fig. S3g–l). Despite that

Harmony, Seurat 3, fastMNN, and Scanorama had high ARI and NMI, and Liger had high iLISI and BatchKL, they all incorrectly split *basal* into two pieces in the UMAP plots (Fig. S3h). scDML was superior in UMAP visualization and ASW_celltype (Fig. S3f, l), and also achieved the best trading-off between batch mixing and clustering accuracy metrics (Fig. S3d, e, j, k).

Then we compared scDML with other methods on a combined human pancreas dataset generated using 5 protocols[36–41], and there are 14,890 cells consisting of 8 batches and 13 cell types, which poses a great challenge due to the strong batch effect. In the UMAP plots of scDML, cells from different batches were well mixed and cells from different cell types were completely separated (Figs. 3a, S5a). scDML also resulted in the cleanest clusters (Fig. S5c), and the highest ASW_celltype, ARI and NMI (Fig. 3b, d), suggesting its ability to improve clustering performance. Even though the batch mixing metrics of Liger and Seurat 3 were higher than scDML (Fig. 3c), they failed to detach some rare cell types in their UMAP embeddings (Fig. S5b). We also displayed some iterations for scDML, and it is clear that scDML converged very quickly during training (Fig. S6).

## scDML identifies subtle cell types

Taking a closer look at the pancreas dataset, both the Sankey plots (Fig. 3e) and clustering plots (Fig. S5b) indicated that clusters detected by scDML agreed well with the pre-annotated labels. Clusters identified by scDML were most consistent with the true cell types, especially for some tiny cell types, while other methods failed to distinguish the tiny clusters. When we highlighted the rare cell types (*activated_stellate, endothelial, epsilon, macrophage, mast, quiescent_stellate, schwann*) in Figs. 3f and S5d, only scDML successfully divided these cell types into isolated clusters.

Similar conclusions can be drawn from the macaque retina dataset, which has multi-level batches with 30,302 cells obtained from 2 different regions, 4 different macaques, and 30 different samples[42]. scDML once again yielded the ideal UMAP visualization (Fig. S7a), superior clustering result (Fig. S7b) and top-ranking metrics (Figs. 3h, S7e, f). Meanwhile, in Figs. 3g and S7c, only scDML succeeded to identify the subtle cell type *OFFx*. It is proved that scDML not only preserved true biological variation, but also especially recovered subtle cell types.

## scDML discovers new clusters when integrating datasets across species

We next set out to test whether scDML can be used to integrate datasets across species. The hypothesis was that the joint analysis can at least preserve all biological information obtained from separated analysis and has the potential to detect new clusters. We downloaded two scRNA-seq datasets from human and mouse lung tissues[43]. Standard preprocessing and dimension reduction show that the two datasets have little overlap (Fig. S8c), indicating serious batch effect. After integration and merging clusters by scDML, a substantial overlap between the two datasets for common cell types was observed (Fig. 4a).

For the integration task, Harmony, Seurat and scDML were better than other methods in terms of ASW_celltype (0.6242, 0.6240 and 0.6239, respectively) (Fig. 4b). Although INSCT achieved the best BatchKL and iLISI index (Fig. 4c), biological differences between the cell types were completely lost given its lowest ARI and NMI (Fig. 4d). The method Liger still performed well on batch mixing (Fig. 4c), but was still inferior to scDML, Seurat 3, Harmony, BERMUDA, scVI, and Scanorama in clustering accuracy (Fig. 4d).

Most importantly, after integration analysis, scDML can discover some rare and tiny subtypes possibly missed by other competing methods. As shown in Figs. 4e, f and S9a, we clearly see that some B cell marker genes (CD79B, JCHAIN, IGHA2, IGHG2, IGHG3, IGHG4) expressed in different clusters for scDML. Some marker genes of

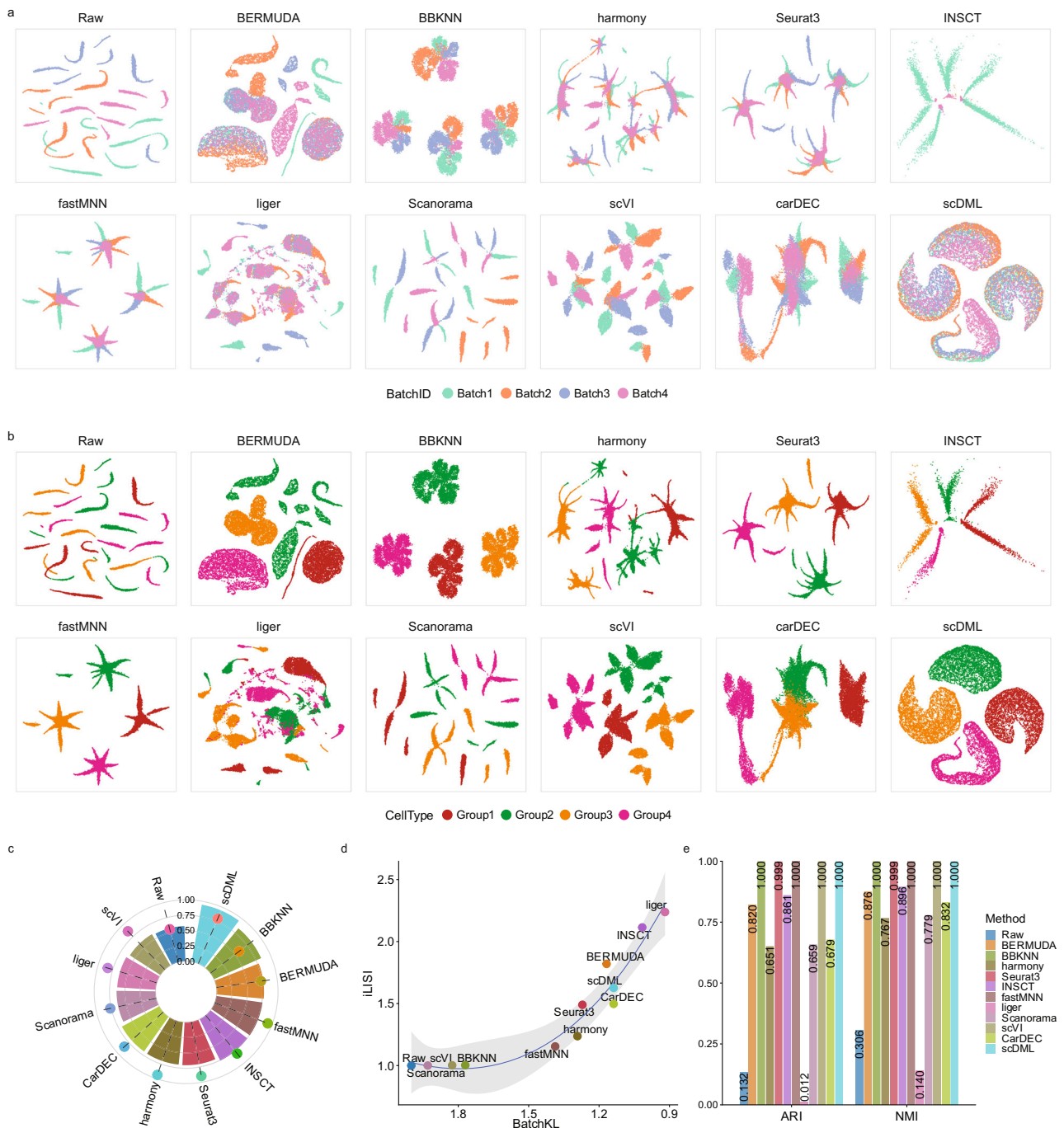

**Fig. 2 | scDML removes batch effects and keeps the biological difference in the complex simulated scRNA-seq data. a** UMAP embedding computed from compared methods, in which the points are colored by batch. **b** UMAP embedding computed from compared methods, in which the points are colored by cell type. **c** Bar plot shows the value of ASW_celltype and ASW_batch, in which the bar height denotes the value of ASW_celltype and the point height denotes the value of ASW_batch. Higher ASW_celltype and lower ASW_batch means better performance. **d** Scatter plot shows the value of BatchKL (x-axis) and iLISI (y-axis). Point closer to the upper right means better performance. The error band means confidence interval of 0.95 level around smooth using B-spline smoothing function with degree equal to 3. **e** Bar plot shows the value of ARI and NMI for different methods. Higher bar means better performance. Source data are provided as a Source Data file.

fibroblast (COL3A1, DCN) and endothelial cells (PRX, MMRN1) also expressed distinctly for scDML. The expression patterns of all these marker genes for scDML are consistent with the cell subtypes annotated in Fig. S8a. However, the most popular methods Seurat 3, Harmony, and scVI failed to detect cell subtypes for B cells (Figs. S9b, S10d), which further proves that scDML can integrate respective characteristics of different species to obtain more refined discoveries.

## scDML is scalable to large-scale dataset

As the scale of scRNA-seq continues to grow, it becomes increasingly important for a method to be scalable to large datasets. To evaluate the scalability of scDML, we analyzed a dataset of 833,206 mouse brain cells, which consists of two batches of murine brain data, acquired using two protocols Drop-seq[44] and SPLiT-seq[45] respectively. This dataset is heavily dominated by the cell type *neuron* (with 560,672 cells), and other cell types are rare. Still,

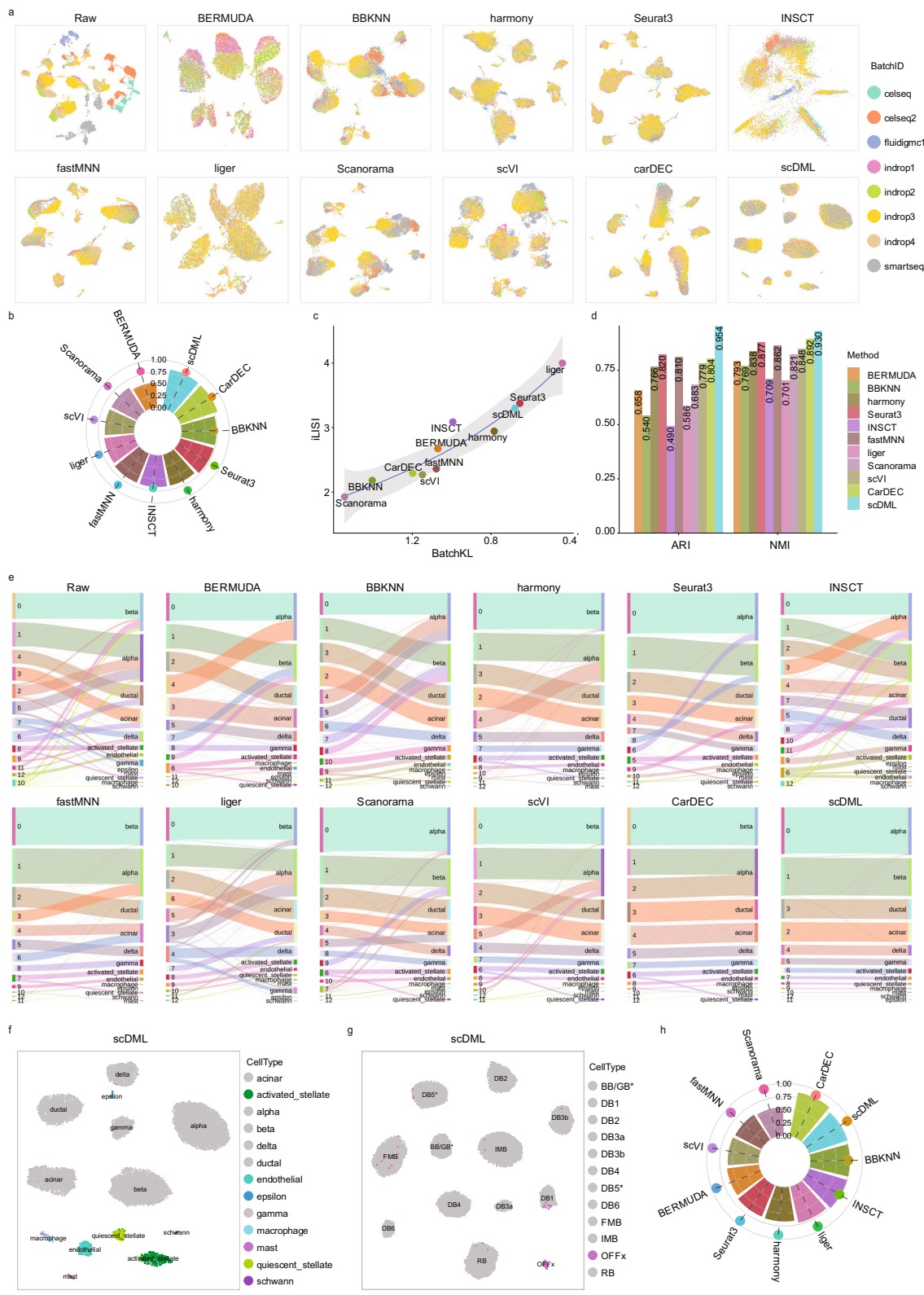

scDML was the best method that separated neurons from other cell types (Fig. 5a, b) and generated the most isolated clusters (Fig. S11a) with the overwhelming highest ARI and NMI (Figs. 5c, S11d). The metrics BatchKL and iLISI of scDML were only inferior to Liger (Fig. S11c), but Liger sacrificed clustering accuracy (Fig. 5c). Other compared methods failed to maintain relatively good cell type separation or batch mixing (Fig. S11b, c). We

excluded Seurat 3, CarDEC and BERMUDA as the compared methods for this dataset because of the huge memory usage and long running time. In most cases, the clustering result of Louvain on learned embeddings is consistent with the reassigned label of scDML, but Louvain is likely to separate a big cluster into several small clusters. For this dataset, compared with Louvain's results, ARI and NMI from the reassigned label of scDML were

**Fig. 3 | scDML accurately preserves rare cell types from original dataset and removes batch effect. a–f** for the human pancreas dataset: (**a**) UMAP embedding computed from compared methods, in which the points are colored by batch. **b** Bar plot shows the value of ASW_celltype and ASW_batch, in which the bar height denotes the value of ASW_celltype and the point height denotes the value of ASW_batch. Higher ASW_celltype and lower ASW_batch means better performance. **c** Scatter plot shows the value of BatchKL (x-axis) and iLISI (y-axis). Point closer to the upper right means better performance. The error band means confidence interval of 0.95 level around smooth using B-spline smoothing function with degree equal to 3. **d** Bar plot shows the value of ARI and NMI for different methods. Higher

bar means better performance. **e** Sankey plot shows the correspondence between the cluster label and the true cell type label. **f** UMAP embedding computed from scDML, and the highlighted cell types are rare cell types. **g, h** for the macaque retina dataset: (**g**) UMAP embedding computed from scDML, and the highlighted cell type *OFFx* is the cell type with the fewest cells. **h** Bar plot shows the value of ASW_cell-type and ASW_batch, in which the bar height denotes the value of ASW_celltype and the point height denotes the value of ASW_batch. Higher ASW_celltype and lower ASW_batch means better performance. Source data are provided as a Source Data file.

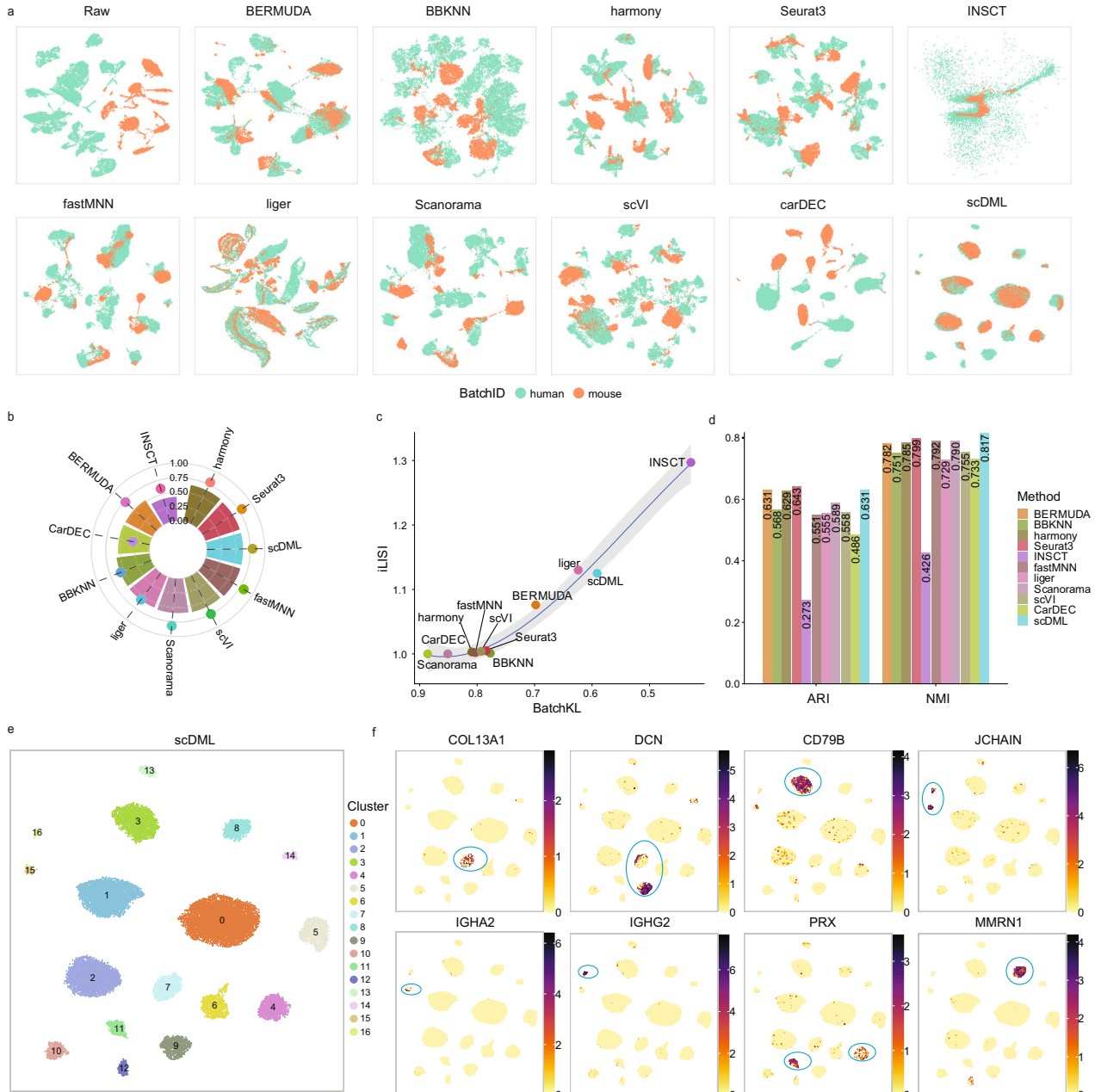

**Fig. 4 | scDML enables cross-species integration for the human and mouse lung datasets. a** UMAP embedding computed from different methods, in which the points are colored by species. **b** Bar plot shows the value of ASW_celltype and ASW_batch, in which the bar height denotes the value of ASW_celltype and the point height denotes the value of ASW_batch. Higher ASW_celltype and lower ASW_batch means better performance. **c** Scatter plot shows the value of BatchKL (x-axis) and iLISI (y-axis). Point closer to the upper right means better performance. The error

band means confidence interval of 0.95 level around smooth using B-spline smoothing function with degree equal to 3. **d** Bar plot shows the value of ARI and NMI for different methods. Higher bar means better performance. **e** UMAP embedding computed from scDML, in which the points are colored by cluster label obtained from scDML. **f** Feature plots show some fibroblast markers (*COL3A1, DCN*), B cell markers (*CD79B, JCHAIN, IGHA2, IGHG2*), and endothelial cell markers (*PRX, MMRN1*). Source data are provided as a Source Data file.

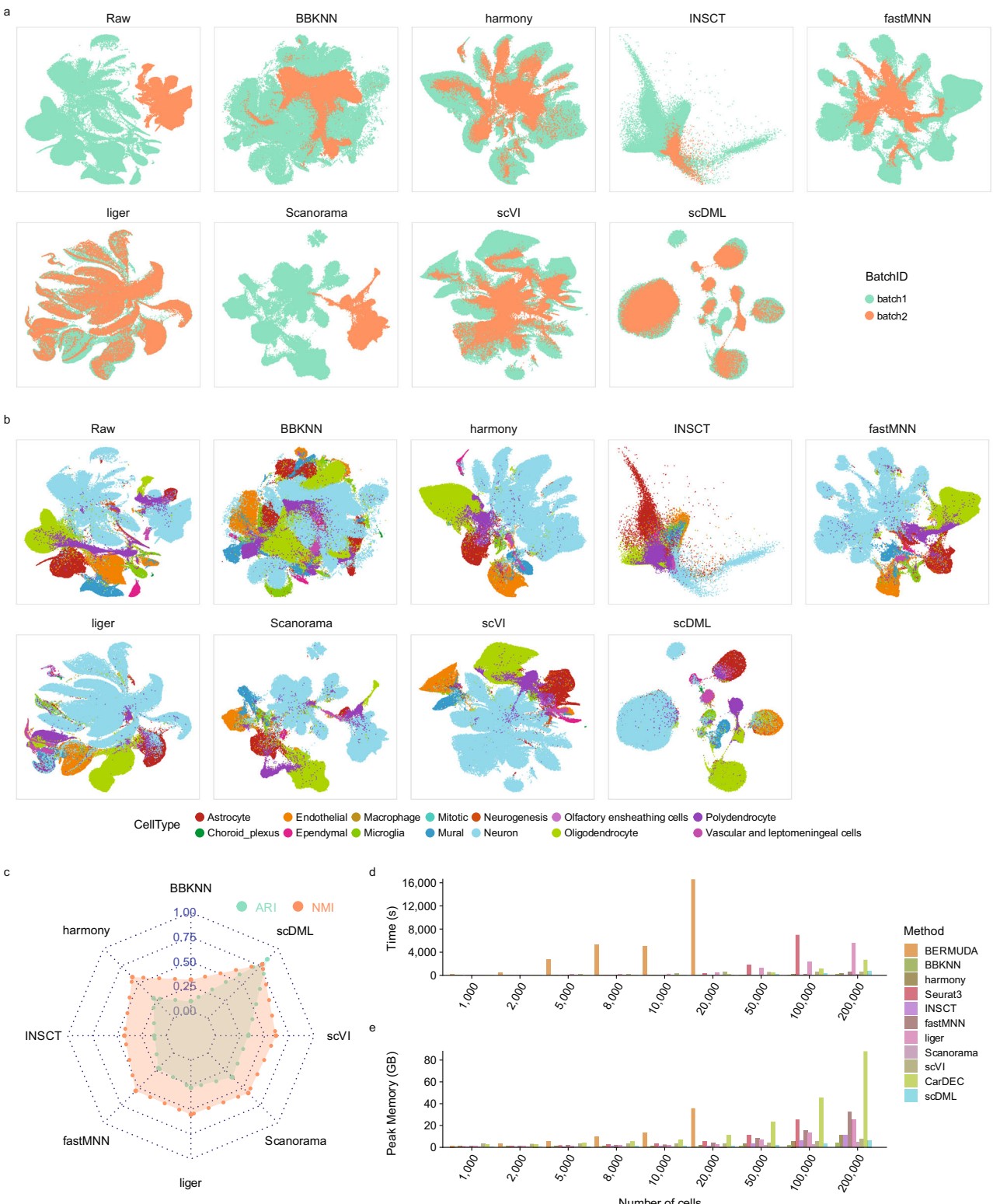

**Fig. 5 | scDML enables integrating large datasets and simultaneously removing batch effect for the mouse brain dataset.** UMAP embedding computed from different methods, in which the points are colored by batch (**a**) and by cell type (**b**). **c** Radar plot shows the value of ARI and NMI for different methods. Higher ARI and NMI means better performance. **d** Bar plot shows the running time of different methods when handling varying number of cells. Lower bar means better performance. **e** Bar plot shows the peak memory usage of different methods when handling varying number of cells. Lower bar means better performance. Remark:

BERMUDA was interrupted early due to the long running time and huge memory requirement when the number of cells ≥ 50,000. Seurat 3 cannot integrate large dataset with the limitation of memory when the number of cells ≥200,000. Additionally, the reported running time and memory usage only include clustering procedure and do not include the procedure of computing t-SNE or UAMP. All reported time and memory usage related to this figure were conducted on Ubuntu 16.04.7 LTS with Intel® Core (TM) E5-2620 v4 CPU @2.10 GHz and 128GB memory. Source data are provided as a Source Data file.

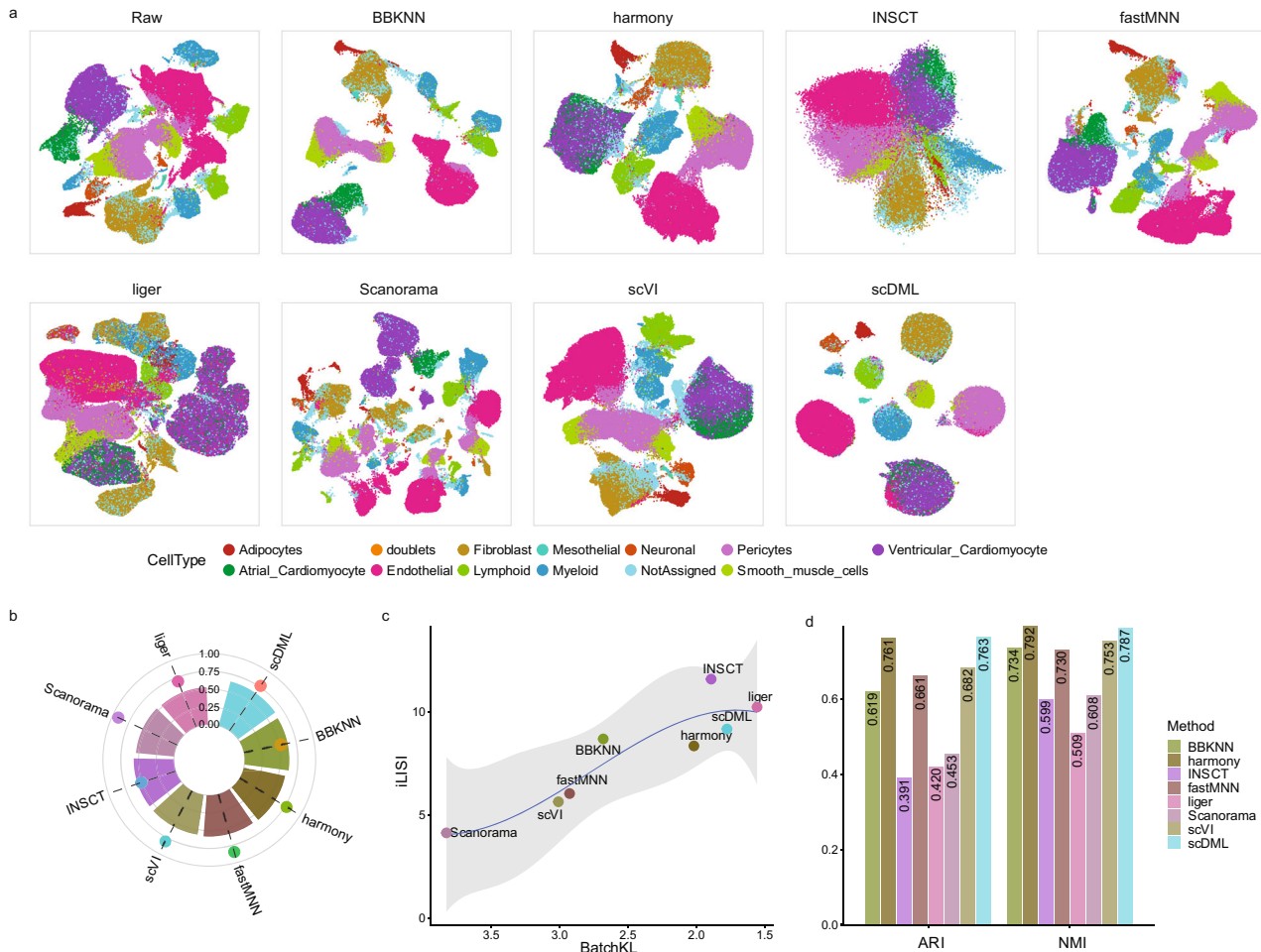

**Fig. 6 | scDML enables integrating large numbers of batches and simultaneously removing batch effect for the healthy human heart dataset with 140 batches. a** UMAP embedding computed from different methods, in which the points are colored by cell type. **b** Bar plot shows the value of ASW_celltype and ASW_batch, in which the bar height denotes the value of ASW_celltype and the point height denotes the value of ASW_batch. Higher ASW_celltype and lower ASW_batch means better performance. **c** Scatter plot shows the value of BatchKL (x-axis) and iLISI (y-axis). Point closer to the upper right means better performance. The error band means confidence interval of 0.95 level around smooth using B-spline smoothing function with degree equal to 3. **d** Bar plot shows the value of ARI and NMI for different methods. Higher bar means better performance. Source data are provided as a Source Data file.

substantially improved (ARI from 0.277 to 0.847, NMI from 0.580 to 0.773), implying that the learned embedding of scDML is biologically meaningful.

Figure 5d, e display the running time and peak memory usage of different methods when processing varying numbers of cells. For a fair comparison. All evaluations were done on Ubuntu 16.04.7 LTS with Intel® Core (TM) E5-2620 v4 CPU @2.10 GHz and 128GB memory and exclude the preprocessing. Although INSCT was the fastest, it sacrificed the clustering accuracy.BBKNN, Scanorama, Harmony, fastMNN, and scVI finished the analysis in less than 15 mins at 20,0000 cells. The running time of scDML was tested on the CPU and the time would be reduced if the GPU is utilized. As for memory usage, scDML ranked third (6.5GB) behind BBKNN and Scanorama, for 3.72 GB and 5.05 GB respectively at 200,000 cells (Fig. 5e). By contrast, BERMUD, CarDEC and Seurat 3 had serious scalability issues. BERMUDA was interrupted early due to the long running time and huge memory requirement when the number of cells was large than 50,000, and Seurat 3 could not integrate large datasets with the limitation of memory when the number of cells was large than 200,000. scDML was computationally fast and memory efficient, making it a desirable tool for the analysis of large-scale single-cell transcriptomics data.

## scDML is able to integrate large numbers of batches

To evaluate the performance of scDML on large samples/batches, we firstly utilized a normal human heart atlas with 485,193 nuclei from 140 batches[46] after quality control. We excluded Seurat 3, CarDEC and BERMUDA as the compared methods for this dataset because of the huge memory usage and long running time. As shown in Figs. 6a, S12a, the raw data has very strong batch effect. Compared with other competing methods, scDML reached the highest ASW_celltype, the highest ARI and the second NMI (Figs. 6b, d and S12). Although Liger and INSCT had the best batch mixing metrics BatchKL and iLISI, they sacrificed clustering accuracy with the worst ARI and NMI (Fig. 6d). The UMAP plot also indicates that scDML is able to detect the tiny clusters, such as *Mesothelial*, which is almost missed by other competing methods (Fig. 6a).

To further demonstrate the ability of integrating multi-level batches, we also analyzed a failing human heart data including 269794 nuclei/cells (220752 nuclei and 49042 cells) with 45 samples from 27 healthy donors and 18 individuals with dilated cardiomyopathy[47]. scDML is able to remove batch effects not only between samples but also between sequencing techniques (Fig. S13a, b, d). Moreover, scDML had the best performance for clustering, reaching the highest ARI and NMI, followed by Harmony (Fig. S13c, f).

## Merge rule of scDML preserves the hierarchical structure of original data

To demonstrate the superiority of the proposed merge rule of scDML, we utilized the mouse retina dataset to explore the detailed merging process. This dataset has 14 cell types across 6 replicates with 23,494 cells[48]. As shown in Figs. 7b, f, S14b, c, compared to other methods, scDML produced 14 clusters that precisely matched true cell type labels, reaching ARI = 0.966, which is slightly inferior to fastMNN (ARI = 0.973) and Scanorama (ARI = 0.972), and the highest NMI = 0.934, followed by fastMNN (ARI = 0.922) and Scanorama (NMI = 0.919). In addition, scDML also achieved desirable batch mixing (Figs. 7d, e and S14a)

The Sankey plots in Fig. 7a show each step in the merge procedure of scDML, as the number of clusters decreased from 16 to 9. The original literature of this dataset provides a hierarchical structure obtained from the true cell type labels[48], which is taken as the gold standard and compared with the hierarchical structure from the output of scDML. Specifically, cell types *BC5C* and *BC5B* were partitioned to two clusters when the number of clusters was 14 (see the 3rd column in Fig. 7a). As the number of clusters decreased, they finally merged into one cluster, remained stable since then, and were no longer merged with other clusters (see the 4th to the last column). Likewise, similar cell types *BC5A* and *BC5D* were traced to merge into cluster 7, and *BC1A* and *BC1B* were traced to merge into cluster 8 in the last column. However, distinct cell types such as *BC8/9* and *BC2* (see cluster 1 and 2 in the last column) were never lost or mixed with other cell types in the merge process, even if they are subtle cell types. The hierarchical structure obtained from scDML is consistent with that from the true labels. Therefore, we conclude that the merge rule of scDML ensures that the hierarchical structure of original data is preserved to the greatest extent (Fig. 7c), and meantime the rarer sub-populations keeps identifiable.

## scDML is robust to varying parameters

Here, we discussed the influence of varying hyper-parameters on the performance of scDML (Supplementary Table 2), including resolution in initial clustering, n_cluster (number of clusters defined finally), K_in (number of neighbors within each batch), K_bw (number of neighbors between batches), HVGs (number of highly variable genes).

Firstly, taking the macaque dataset as example, we chose three different resolutions for Louvain algorithm (3.0, 6.0, 9.0), two different numbers of highly variable genes (1000 and 2000), three numbers of clusters after merging (11, 12 and 13), and 21 different combinations for (K_bw, K_in). We used both ARI and NMI to measure the performance of scDML. It is apparent in Fig. 8a, b that scDML is very robust to these hyper-parameters, where both ARI and NMI are greater than 0.9 in all cases.

Taking the pancreas dataset as example, we respectively selected Louvain[49] and Leiden[50] as the initial clustering method, successively increased the resolution from 2.0 to 9.0 (the number of cluster ranges from 30 to 100 accordingly), and computed both ARI and NMI. It is apparent in Fig. 8c, d that the number of clusters increased with resolutions, but the classification accuracy metrics hardly fluctuated, and was also not affected by the initial clustering method. scDML always maintained high ARI and NMI close to 1. Consequently, scDML is robust enough to the varying hyper-parameters.

## Discussion

Many techniques, tools, and platforms for scRNA-seq are already applicable for comparisons across different tissues, disease status, or different species. These diverse datasets necessitate methodologies that can reconcile the technical and biological batch effects inherent in single-cell sequencing technologies. So, in this article, we developed scDML, a method that integrates multiple scRNA-seq datasets to detect potential novel clusters and remove batch effect

simultaneously. scDML has been extensively tested using simulated and real datasets from different species (human, macaque, mouse) and tissues (pancreas, retina, brain, lung) generated by different scRNA-seq protocols. We note that scDML achieved competitive clustering accuracy and batch effect removal compared to current state-of-the-art integration methods for scRNA-seq, such as Seurat 3, fastMNN, Harmony, scVI, BERMUDA, Liger, Scanorama, BBKNN, and INSCT. Moreover, as for scalability and memory usage, scDML outperformed most competing methods. Because scDML is developed based on the framework of PyTorch, it can take advantage of GPU to speed up computation when available.

We proposed a strategy to merge initial clusters successively that takes batch effect into consideration, by computing the number of KNN pairs intra batch and MNN pairs inter batches, then calculating the similarity of clusters, and finally constructing a hierarchical tree, in which the root of the tree is the unique cluster obtained after gathering all clusters, and the leaves are the clusters to be merged. Thereafter, we used the above MNNs to guide information for building better low-dimensional embeddings. In this way, this procedure guarantees that scDML outperforms existing methods in terms of merging the same cell types, separating different cell types and preserving cell types unique to some batches. As for how many clusters should be finally defined, we provided a strategy to automatically infer the number of clusters with eigenvalues of the similarity matrix inspired by spectral clustering (Algorithm 3 in Supplementary Note) or manually set the number of clusters based on the heatmap of similarity matrix. Supplementary Table 4 lists the suggested number of clusters based on above algorithm. Figure S15a, b indicates that the number of clusters for *bct* and *bct_del* datasets should be set to 3, which is the same as the recommendation of Algorithm 3. As for lung dataset and macaque retina dataset, Algorithm 3 suggested the numbers of clusters to be [5,10,13,16,18] and [8,12,18], respectively. In practice, we can also manually merge the similar clusters based on marker genes.

Most importantly, a remarkable feature of scDML is that it not only preserves the rare cell type information of the original dataset, but also has the potential to discover rare clusters that might be hard to extract by analyzing each batch individually as in other competing methods. Additionally, scDML can recover the hierarchical structure underlying the data that has been mostly ignored in the compared methods. What's more, scDML is scalable to large datasets, able to handle multi-level-batch datasets and robust to varying hyper parameters. Therefore, we believe that scDML will be a valuable tool for biomedical researchers to better disentangle complex cellular heterogeneity.

Additionally, the merging rule we proposed is not only applicable to datasets with multiple batches but also improves the performance on datasets with only a single batch. We compared scDML with two commonly used clustering methods, Kmeans and Louvain, on three different datasets. Then we are surprised to find that both ARI and NMI were improved by scDML relative to Kmeans and Louvain (Fig. S16), which also proves the effectiveness and rationality of our proposed method.

One limitation of our method is that scDML can be applied to scRNA-seq datasets with categorial structures, but not those with differentiated structures. Besides, like most batch effect removal methods, scDML only creates an integrated low-dimensional embedding and does not provide corrected gene expression like CarDEC. In the future, we plan to extend the application of scDML to remove batch effect for scRNA-seq at the gene expression level directly, so as to conduct downstream differential expression analysis.

In summary, extensive benchmarking of real datasets and simulated datasets suggests that scDML not only better recovers biological difference and removes batch effect, but also can preserve the rare cell type structure and identify novel cell types that might be ignored by separate analysis. Therefore, we anticipate that scDML will be a

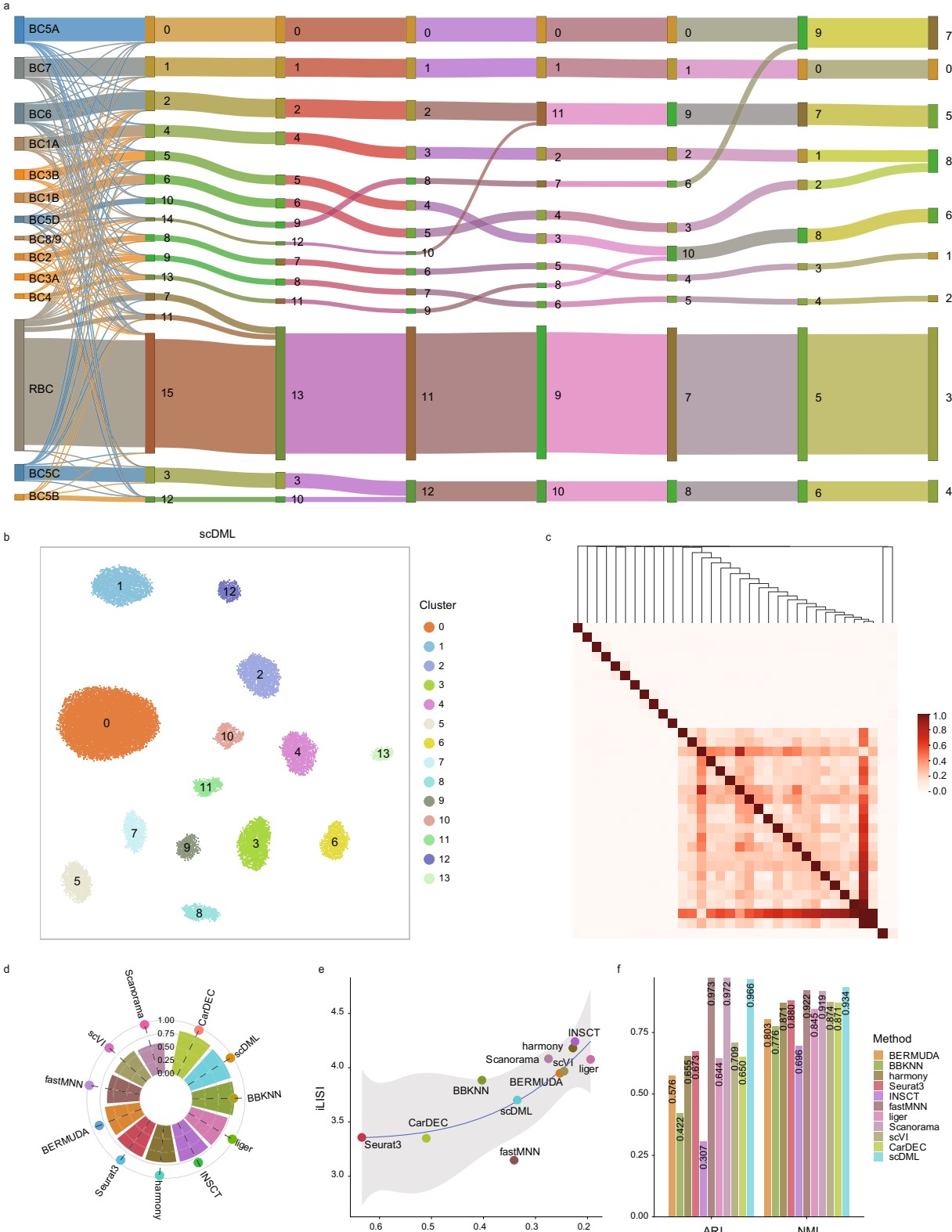

**Fig. 7 | scDML preserves the hierarchical structure when merging clusters for the mouse retina dataset. a** Sankey plot shows the merge procedure for scDML. Columns from left to right respectively denote true cell type labels, cluster labels when the number of clusters equals 16, 14, 13, 12, 11, 10 and 9. **b** UMAP embedding computed from scDML, in which the points are colored by cluster label when the number of clusters is set to 14. **c** Heatmap shows the hierarchical merging order. **d** Bar plot shows the value of ASW_celltype and ASW_batch, in which the bar height denotes the value of ASW_celltype and the point height denotes the value of ASW_batch. Higher ASW_celltype and lower ASW_batch means better performance. **e** Scatter plot shows the value of BatchKL (x-axis) and iLISI (y-axis). Point closer to the upper right means better performance. The error band means 0.95 level of confidence interval around smooth using B-spline smoothing function with degree equals to 3. **f** Bar plot shows the value of ARI and NMI for different methods. Higher bar means better performance. Source data are provided as a Source Data file.

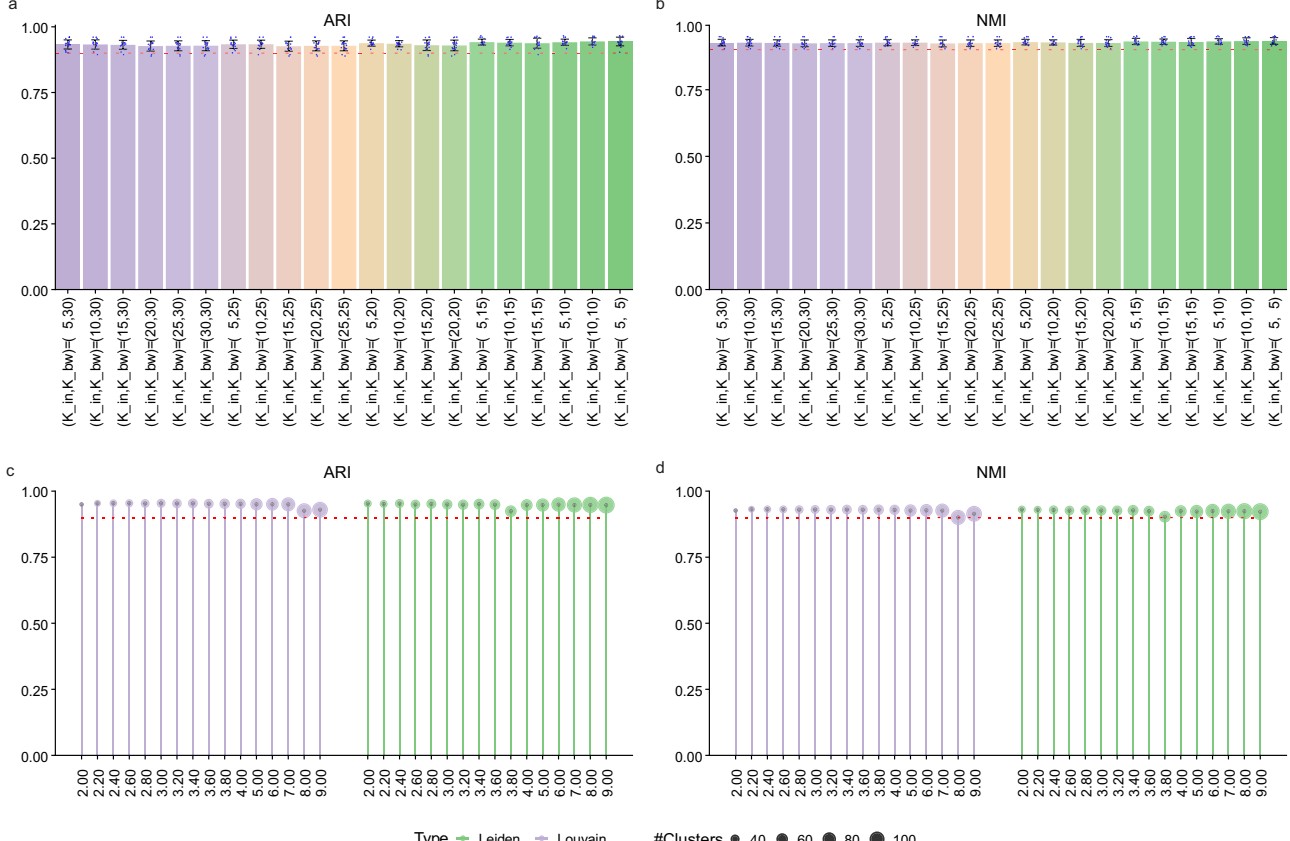

**Fig. 8 | Robustness of scDML to varying hyper parameters.** The ARI (**a**) and NMI (**b**) for scDML with three different resolutions for Louvain algorithm (3.0, 6.0, 9.0), two different numbers of highly variable genes (1000 and 2000), three numbers of clusters after merging (11, 12 and 13), and 21 different combinations for (K_bw, K_in). Each bar represents a combination of (K_bw, K_in). There are 18 cases with respect to each bar, where the bar height is the mean value and the error bar is the standard deviation (Data are presented as mean values ± SEM). The ARI (**c**) and NMI (**d**), respectively using Louvain and Leiden as the initial clustering method by fixing (K_bw, K_in) = (10, 5). The resolution is increased from 2.0 to 9.0. the point size means the number of initial clusters and the point height means the value of ARI and NMI. The red dashed line is the 0.9 level line for reference. Source data are provided as a Source Data file.

valuable tool for the comprehensive analysis of multiple scRNA-seq datasets. Finally, all analysis script described in this manuscript is available via https://github.com/eleozzr/scDML_reproduce to reproduce the results and figures.

## Methods

The scDML workflow (Fig. 1) mainly involves five steps, which are preprocessing, initializing clusters based on PC embeddings, finding NN pairs in the PCA embedding space, constructing the similarity matrix between clusters, and deep metric learning.

### Step1: preprocessing

There are five important tasks to be completed in preprocessing: filtering low-quality cells and genes, cell normalization, log normalization, detecting highly variable genes (HVGs), and z-score normalization with truncated values. All the above steps are implemented in the python module *scanpy*[26] with version 1.7.2.

Let **X** be an $n \times p$ matrix of scRNA-seq data, with $n$ cells, $p$ genes and $M$ batches, and let $x_{ij}$ be the expression value of gene $j$ in cell $i$. In the filtering step, we first remove low quality cells with nGene <10 and remove genes with nCells <3. In cell normalization, we divide counts for each cell by the total counts over all genes and multiply a constant 10000 using *sc.pp.normalize_total* function, to obtain the normalized expression $y_{ij}$. We then conduct log transformation for $y_{ij}$ using *sc.pp.log1p* function. After that, we select highly variable genes by using *sc.pp.highly_variable_genes* and then scale data using *sc.pp.scale* within each batch.

### Step2: initializing clusters in PCA embedding space

Let $\mathbf{Y}_{HVG}$ be the $n \times p_{HVG}$ matrix of normalized expression from **Step 1**, including only the $p_{HVG}$ highly variable genes. Let $y_{i,HVG}$ be the expression value of HVGs in cell $i$, i.e., the $i$th row of $\mathbf{Y}_{HVG}$. To get a suitable initial clustering result for scDML, we firstly conduct PCA (Principal Component Analysis) on $\mathbf{Y}_{HVG}$ to get a low-dimensional embedding $\mathbf{X}_{emb}$. Unless otherwise stated, we set the number of principal components $n_{pca}$ to be 100 without losing too much information. It is easily implemented in *scanpy* package with the function *sc.tl.pca* (*n_components* = 100). Then scDML applies the Louvain method, a graph-based clustering method, on the reduced PCA embedding space to get an initialized clustering result. This procedure can be implemented by the function *sc.tl.louvain* in *scanpy* package, higher resolution means finding more and smaller clusters. However, the number of true cell types is usually unknown in real data, so how to find the right resolution for the Louvain algorithm is a challenge. scDML set a relatively large resolution (3.0 by default) in the Louvain algorithm, which may help to find more subtle cell types in datasets. To remove batch effect in the dataset, it is natural to merge similar clusters intra and inter batches.

### Step3: finding NN pairs in PCA embedding space

To merge the initialized clusters obtained from the Louvain algorithm in **Step2**, we need to compute the similarity between the clusters. Similar to Conos algorithm[51], scDML firstly builds a joint graph between all clusters by finding NN (nearest neighbor) pairs intra batch and inter batches.

## Finding KNN pairs intra batch

Let $\mathbf{X}_{emb} = (\mathbf{X^1},...,\mathbf{X^M})^T$ be a $n \times n_{pca}$ matrix of scRNA-seq data in PCA embedding space, where $\mathbf{X^k}(k=1,2,\cdots,M)$ is $n_k \times n_{pca}$ submatrix of cells in the $k^{th}$ batch. Let $x_i^k$ be the vector of cell $i$ of batch $k$ in PCA embedding space, that is, the $i^{th}$ row of $X^k$. Denote $S_k$ as the set of all KNN pairs in batch $k$, cell $i$ and cell $j$ form a KNN pair if and only if

$$(i,j) \in S_k \, and \, (j,i) \in S_k \Longleftrightarrow \begin{cases} i,j \in B_k \\ i \in \text{KNN}\left(x_j^k\right) or \, j \in \text{KNN}(x_i^k) \end{cases} \quad (1)$$

where the tuples $(i,j)$ and $(j,i)$ are both KNN pairs, $B_k$ is the set of cells belonging to the batch $k$. For the clarity of the following description, we treat $(i,j)$ and $(j,i)$ as different NN pairs. $KNN\left(x_j^k\right)$ represents the set of k-nearest neighbors of cell $i$ in batch $k$. To find KNNs intra batches, we set the number of neighbors $K_{in} = 5$ by default and uses cosine distance for scDML.

## Finding MNN pairs inter batches

Many studies have proved that MNN (mutual nearest neighbor) based methods can effectively remove batch effect in scRNA-seq data, such as MNN[8,9], BEER[52], BBKNN[11], Scanorama[10] and INSCT[20]. So, here we also use the MNN pairs to construct the similarity of clusters between different batches. To correspond with the definition of KNN pairs intra batch, let $S_{a,b}$ be the set of MNN pairs between batch $a$ and batch $b$, then cell $i$ and cell $j$ form an MNN pair if and only if

$$(i,j) \in S_{a,b} \, and \, (j,i) \in S_{a,b} \Longleftrightarrow \begin{cases} i \in B_a, j \in B_b \\ i \in \text{MNN}\left(x_j^b, X^a\right) and \, j \in \text{MNN}\left(x_i^a, X^b\right) \end{cases}, \quad (2)$$

where the tuples $(i,j)$ and $(j,i)$ are both MNN pairs, $\text{MNN}\left(x_i^a, X^b\right)$ represents the set of cells in batch $b$ which are nearest to cell $i$ in batch $a$, and $\text{MNN}\left(x_j^b, X^a\right)$ represents the set of cells in batch $a$ which are nearest to cell $j$ in batch $b$. scDML sets the number of neighbors $K_{bw} = 10$ by default and uses cosine distance to calculate MNN pairs.

## Step4: calculate similarity of clusters and construct hierarchical cluster tree

Let $S_{in}$ represents all KNN pairs intra batches, $S_{bw}$ represents all MNN pair inter batches and $S$ represents all NN pairs in the dataset.

$$S_{in} = S_1 \cup S_2 \cup \cdots \cup S_M \quad (3)$$

$$S_{bw} = \bigcup_{j=1}^{M} \bigcup_{j=i+1}^{M} S_{ij} \quad (4)$$

$$S = S_{in} \cup S_{bw} \quad (5)$$

Let $N = |S|$ denote the total number of NN pairs in set $S$. Based on the clustering results in Step2 and all NN pairs obtained in Step 3, we calculate the number of NN pairs between pairwise clusters, and define a symmetric matrix $A$ as

$$A = \begin{bmatrix} a_{1,1} & a_{1,2} & \cdots & a_{1,C} \\ a_{2,1} & a_{2,2} & \cdots & a_{2,C} \\ \vdots & \vdots & \ddots & \vdots \\ a_{C,1} & a_{C,2} & \cdots & a_{C,C} \end{bmatrix}, \quad (6)$$

where $a_{i,j}$ represents the number of NN pairs between cluster $i$ and cluster $j$. It is worth noting that scDML deletes all the NN pairs (kNN and MNN pairs) that belong to the same cluster, that is, we set the elements on the diagonal of $A$ to 0. Obviously, the smaller the cluster size is, the

less the number of NN pairs will be found. So, we should take the cluster size (the number of cells in each cluster) into consideration when using $A$ to represent the similarity (or connectivity) between clusters. To achieve the above goals, scDML adopts a simple but intuitive method to calculate the similarity matrix between clusters,

$$\mathbf{S} = \begin{bmatrix} s_{1,1} & s_{1,2} & \cdots & s_{1,C} \\ s_{2,1} & s_{2,2} & \cdots & s_{2,C} \\ \vdots & \vdots & \ddots & \vdots \\ s_{C,1} & s_{C,2} & \cdots & s_{C,C} \end{bmatrix}, \quad (7)$$

$$s_{i,j} = \frac{a_{i,j}}{\min(m_i, m_j)}, i,j = 1,...,C, \quad (8)$$

where $m_i$ denotes the number of cells in cluster $i$, and the matrix $S$ is still symmetric. Inspired by Lihi et al.[53], we provided a strategy to automatically infer the number of clusters according to the eigenvalues of the similarity matrix $\mathbf{S}$ (Algorithm 3 in Supplementary Note).

In particular, scDML applies a merging rule different from BERMUDA, which can generate a more stable result. scDML utilizes the idea of hierarchical clustering to merge clusters one by one. For the detailed merging procedure please refer to Algorithm 1 in Supplementary Note.

After the above merging procedure, we finally obtain a set $P$ whose element can be viewed as an edge in an undirected graph $G$, in which the node of $G$ is the clustered index $\{1, ..., C\}$. Then we can find all connected components in graph $G$ to reassign the label of cluster. Suppose that we have obtained $K$ connected components in graph $G$. We represent each connected component as $G_i$, $i = 1, ..., K$, where each $G_i$ is the subset of $\{1, ..., C\}$ and they are disjoint. That is to say,

$$\bigcup_{i=1}^{K} G_i = \{1, \ldots, C\}, G_i \cap G_j = \varnothing, \forall i \neq j, i = 1, \cdots K, j = 1, \cdots K. \quad (9)$$

$K$ can be set as the final number of expected clusters. In other words, all clusters belonging to a same connected component should be considered as one cluster so that we can merge the initialized clusters to the updated clusters, taking the batch effect into account.

## Step5: deep metric learning to remove batch effect

As we know, MNN pairs can help to remove batch effect to some degree. Therefore, we tend to use the above MNN-guided information to build better low-dimensional embeddings. In addition, metric learning is an approach based on distance metric directly, which aims at automatically constructing task-specific from (weakly) supervised data[54,55]. Although scDML has reassigned the cluster label of the dataset, raw data has not been corrected for batch effect actually. Here we use the deep metric learning (DML) method with triplet loss to capture more accurate low-dimensional representation. Broadly speaking, our goal is to learn a distance metric that pulls points with the same label close together while pushing away points with different labels, meanwhile considering the influence of batch effect.

## Triplet definition

In step 4, we have obtained the cluster label for each cell. To make full use of the cluster information, we construct triples (anchor, positive, negative) according to the following guidelines. Given a cell $a$ (anchor point), we randomly choose a cell $p$ as a positive point from cells whose cluster label is the same as $a$ and randomly select a cell $n$ as a negative point from cells whose cluster label is different from $a$. The tuple $(a, p, n)$ can be regarded as a triplet. Any cell in the dataset can be used as an anchor.

## Triplet loss

Here, the triplet loss function is defined as follows:

$$L(a, p, n) = \max(d(a, p) - d(a, n) + m, 0), \qquad (10)$$

where $d$ is the distance metric and we adopt Euclidean distance. $m$ is a margin between similar and dissimilar pairs, by default $m = 0.2$. The triplet loss is optimized by minimizing the distance between anchor-positive pairs and maximizing the distance between anchor-negative pairs. Based on the definition of the triplet loss, there are three possible categories of triplets:

1. **Easy triplets**: triplets which have a loss of 0, that is, $d(a, p) + m < d(a, n)$;
2. **Hard triplets**: triplets where the negative point is closer to the anchor point than the positive point, i.e., $d(a, n) < d(a, p)$;
3. **Semi-hard triplets**: triplets where the negative point is not closer to the anchor point than the positive point, but the loss is still positive. i.e. $d(a, p) < d(a, n) < d(a, p) + m$.

As can be seen from above, easy triplets do not affect the optimization of triplet loss. Semi-hard triplets can be used for the optimization, but it will find too many triplets, which will cost much training time and memory for real datasets. Considering the time and memory consumption, we select hard triplets found in embedding space to train the scDML model.

## The structure and training of scDML

Unless otherwise specified, scDML adopts a simple embedding network with the number of nodes of input layer, the hidden layer and the embedding layer being 1000, 256, and 32 respectively. According to the definition of triplets, the anchors are independent, and thus we can optimize the triplet loss by a mini-batch strategy. For detailed training procedure please refer to Algorithm 2 in Supplementary Note 1, where $f(\cdot)$ represents the nonlinear function mapping input to the embedding net. The implementation of scDML is based on the PyTorch framework, which makes full use of a scalable package named *pytorch_metric_learning*.

## Evaluation metrics

To benchmark the various competing methods on different datasets, the following six evaluation metrics are employed to quantify the concordance of clustering results and the ability of removing batch effect.

**ARI.** Adjusted rand index (ARI) is used to quantify clustering accuracy, and can be calculated by the function *adjusted_rand_score* in the python module *sklearn.metrics.cluster*. ARI measures the similarity between two clustering results, defined as

$$\text{ARI} = \frac{\sum_{ij} \binom{n_{ij}}{2} - \left[ \sum_i \binom{a_i}{2} \sum_j \binom{b_j}{2} \right] / \binom{n}{2}}{\frac{1}{2} \left[ \sum_i \binom{a_i}{2} + \sum_j \binom{b_j}{2} \right] - \left[ \sum_i \binom{a_i}{2} + \sum_j \binom{b_j}{2} \right] / \binom{n}{2}}, \qquad (11)$$

where $n_{ij}$ is the number of cells in both cluster $i$ of the clustering result and cell type $j$ of the true cell type labels, $a_i$ is the number of cells from cluster $i$, $b_j$ is the number of cells from cell type $j$, and $n$ is the total number of cells. We calculate ARI to compare the clustering result of integrated data with the predefined cell types. ARI ranges in [0, 1], and higher values indicate higher similarities.

**NMI.** Normalized mutual information (NMI) is also used to measure clustering accuracy, and can be calculated by the function *normalized_mutual_info_score* in the python module *sklearn.metrics.cluster*.

NMI is defined as

$$\text{NMI} = 2 \times \frac{\sum_{ij} \frac{n_{ij}}{n} \log\left( \frac{n \times n_{ij}}{a_i \times b_j} \right)}{\sum_i \frac{a_i}{n} \log\left( \frac{n}{a_i} \right) + \sum_j \frac{b_j}{n} \log\left( \frac{n}{b_j} \right)}, \qquad (12)$$

where the notations are the same as that in ARI. NMI ranges in [0, 1] and higher values also indicate higher similarities between the clustering result and true cell types.

**ASW_celltype.** Average silhouette width for cell type (ASW_celltype) is used to assess cell type purity, and can be calculated by the function *silhouette_score* in the python module *sklearn.metrics*. The silhouette width for cell type label of cell $i$ is defined as

$$s_i = \frac{b_i - a_i}{\max\{a_i, b_i\}}, \qquad (13)$$

where $a_i$ is the average distance from cell $i$ to all cells with the same label, and $b_i$ is the lowest average distance of cell $i$ to each group of cells which are assigned different labels. ASW_celltype is the mean of silhouette widths across all cells and ranges in [0, 1], where higher values indicate cells are closer to cells with the same label and further away from cells with a different label. We calculate ASW_celltype based on predefined cell type labels in the low-dimensional (with dimensions=32 by default) PCA embedding space.

**ASW_batch.** Average silhouette width for batch (ASW_batch) is used to evaluate how well batches are globally mixed. The silhouette width for batch of cell $i$ is defined as

$$s_i = \frac{b_i - a_i}{\max\{a_i, b_i\}}, \qquad (14)$$

where $a_i$ is the average distance from cell $i$ to all cells of the same batch, and $b_i$ is the lowest average distance of cell $i$ to each group of cells which are assigned to other batches. ASW_batch is the mean of silhouette widths across all cells, where higher values indicate cells are closer to cells of the same batch and further away from cells of a different batch. We inferred that higher ASW_batch means that batches are more mutually exclusive, and conversely, lower ASW_batch typically indicates better batch mixing and batch effect correction. ASW_batch also range in [0, 1].

**iLISI.** Inverse Simpson's index of integration (iLISI) is used to evaluate how well batches are locally mixed after integration. The local inverse Simpson's index (LISI) can be used to measure the batch distribution (iLISI), based on local neighbors chosen on a preselected perplexity. Using the selected neighbors of a cell, the LISI was then computed on the batch labels for the iLISI index, and a score close to the expected number of batches denotes good batch mixing. We compute iLISI using the function *compute_lisi* in the R package *lisi* for all cells in the dataset, output the average score and higher iLISI indicates better performance for batch mixing.

**BatchKL.** KL divergence is used to evaluate the performance of methods in batch effect removal based on the embedding representation. For the definition of KL divergence of batch mixing (BatchKL) please refer to Li et al.[31]. The lower value denotes the better mixing performance.

## Reporting summary

Further information on research design is available in the Nature Portfolio Reporting Summary linked to this article.

## Data availability

We analyzed multiple published scRNA-seq datasets and two simulated datasets, which are available through the accession numbers reported in the original articles. (1) Simulated datasets: generated by splatter from Luecken et al.[17], which can be accessed by https://figshare.com/articles/dataset/Benchmarking_atlas-level_data_integration_in_single-cell_genomics_-_integration_task_datasets_Immune_and_pancreas_/12420968(sim1_1_norm.h5ad, sim2_2_norm.h5ad); (2) Mammary epithelial datasets: mammary epithelial cells from three independent studies, and can be downloaded from https://doi.org/10.6084/m9.figshare.20499630.v2[56]; (3) Human pancreas dataset: We used a pre-annotated collection from the tutorial of Seurat (https://satijalab.org/seurat/archive/v3.2/integration.html, standard workflow) with accession codes "GSE81076", "GSE85241", "GSE86469", "GSE84133" and "E-MTAB-5061 []"; (4) Macaque retina dataset: "GSE118480"; (5) Mouse retina dataset: "GSE81904"; (6) Mouse brain datasets: "GSE116470" and "GSE110823", which can be also downloaded from http://scanorama.csail.mit.edu/data.tar.gz; (7) Human lung and mouse lung dataset: "GSE133747"; (8) Healthy human heart dataset: https://www.heartcellatlas.org/, which can be downloaded from https://doi.org/10.6084/m9.figshare.20499630.v2[56]; (9) Failing human heart dataset with multiple-level-batch: "GSE183852"; (10) Single batch datasets: three datasets analyzed are processed by Chen et al.[57] and can be downloaded from https://drive.google.com/drive/folders/1BIZxZNbouPtGf_cyu7vM44G5EcbxECeu (Adam, Muraro and Quake_10X_Limb_Muscle). Details of these datasets are described in Supplementary Data 1. All datasets analyzed are available from https://doi.org/10.6084/m9.figshare.20499630.v2[56]. All other relevant data supporting the key findings of this study are available within the article and its Supplementary Information files. Source data are provided with this paper.

## Code availability

scDML algorithm is implemented in python based on the PyTorch framework and avaliable via https://github.com/eleozzr/scDML[58]. All analyses and results presented in the manuscript are available via https://github.com/eleozzr/scDML_reproduce. scDML is licensed under the MIT license.

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

## Acknowledgements

This work was supported by Changping Laboratory and China Postdoctoral Science Foundation funded project (to X.L.), Public Health & Disease Control and Prevention, Fund for Building World-Class Universities (Disciplines) of Renmin University of China (to J.Z.) and the disciplinary funding of Central University of Finance and Economics (to X.X.).

## Author contributions

This study was conceived of and led by X.L. and J.Z.. X.Y. designed the model and algorithm, and implemented the scDML software. X.Y. led the data analysis with input from X.L.. X.X., X.Y. and X.L. wrote the paper with feedback from all other coauthors. All authors read and approved the manuscript.

## Competing interests

The authors declare no competing interests.
