## [Peer Review File · Nature Communications]

Batch alignment of single-cell transcriptomics data using deep metric learningReviewer #1 (Remarks to the Author):

Yu et al. presented a method named scDML for batch integration of single-cell transcriptomics data. The method is guided by the initial clusters and the nearest neighbors within and between batches. Using simulation and experimentally acquired datasets, the authors demonstrated that scDML outperformed competing methods.

Major concerns:

1. I am concerned that methods such as BBKNN, Harmony, Seurat3, scVI were not run properly or not run in their optimal ways. These methods are very popular and widely adopted by researchers. Several benchmark studies also demonstrated their superior performance on a wide range of applications. However, in Figure 2, Figure 4, Figure 5, the authors showed that these four methods performed poorly. Much better performance is expected from these methods.
2. Is scDML able to integrate large numbers of batches/samples? For instance, constructing a covid atlas from more than 1000 samples/patients. What is the upper limit of number of batches/sample?
3. In most of the figures, UMAP plots before batch integration should be added to show the presence of batch effects.
4. In Figure 3, UMAP plots should be shown the main figure, just like Figure 2, 4, 5.
5. The Sankey plots in Figure 3A are misleading. The authors stated that there are much more crossings in the Sankey plots of competing methods (Line 172). These crossings can be easily removed by reordering the clusters. For instance, in the harmony plot, if we move cluster 4 to between 1 and 3, the Sankey plot will have less crossings.

Minor comments:

1. Line 98, reference is required for BERMUDA.

Reviewer #2 (Remarks to the Author):

In "Batch alignment of single-cell transcriptomics data using deep metric learning," Yu et al. propose a new method for scRNA-seq integration using a multistep pipeline involving clustering, MNN search to find anchors, and deep metric learning using a small neural network with triplet loss to find an integrated embedding. Overall, I found the method sound and interesting in principle, with a good amount of benchmarking using appropriate metrics, and that shows the method performs comparably well against existing tools. I therefore recommend publication of the manuscript after minor revisions, which are noted below

- What are the noteworthy results?

The process of high-resolution clustering, finding MNNs based on these clusters (and potentially merging them), followed by triplet loss is a good approach. The benchmarking results look promising.

- Will the work be of significance to the field and related fields? How does it compare to the established literature? If the work is not original, please provide relevant references.

The method could be another useful one to try among the current suite of integration tools. The authors provide good comparison to existing literature. The method makes use of existing approaches, e.g., Haghverdi et al. (2018) introduced MNNs, clustering algorithms like Louvain are common in scRNA-seq, triplet loss and metric learning has been used in a number of studies like Simon et al., 2021, though the particular pipeline and implementation are original.

- Does the work support the conclusions and claims, or is additional evidence needed?

Yes

- Are there any flaws in the data analysis, interpretation and conclusions? - Do these prohibit publication or require revision?

No

- Is the methodology sound? Does the work meet the expected standards in your field?

Yes

- Is there enough detail provided in the methods for the work to be reproduced?
Authors do mostly a good job, a few points noted below.

Other comments

1. Page 1, line 20: "unappreciated levels of heterogeneity": This is a vague statement and the sentence could just be deleted.
2. Line 21: "accurately detecting the number of cell types": This problem is not rigorously addressed by the authors and so this phrase should be considered for removal.
3. Page 3, lines 56-7: My understanding of the benchmarking study of Luecken et al. (reference 12) is that the neural architectures (scVI, SCANVI) perform best overall, especially for major dataset differences, with Scanorama performing better for subtle differences. I am also not sure what "over-denoised" means in line 64. The authors should comment on this.
4. Line 72: "make the clusters as many as possible": The authors should just say something like "clustered at a high resolution (Louvain resolution of 3.0)" or something to that effect. As is, this is not an accurate statement (i.e., maximizing the clusters would theoretically just put each cell into its own cluster).
5. Page 4, line 78: "serval" should be "several"
6. Page 11, line 248: "significantly" should be "substantially" to avoid confusion with statistical significance
7. Page 22, lines 479-486 and Algorithm 2: I was unable to find important details on the architecture of the embedding network. Aside from the loss and the numbers of nodes per layer, which are provided, the authors should also provide the activation functions, connectivity structure, etc.

Reviewer #3 (Remarks to the Author):

In this manuscript, Yu et al. presented a deep metric learning-based method scDML for correcting the batch effect among single-cell RNA-seq data. scDML borrows the initial cluster information to refine the clustering and remove batch effect. According to the evaluation results, scDML outperformed the state-of-the-arts methods, such as Seurat, Harmony, scVI etc. However, there are still some concerns that have to be addressed before any further consideration.

Major comments:

Now methods performing batch effect with the help of cell cluster information are now something really new. For example, SMNN and iSMNN by Yang et al. (2021; 2021) both take the advantage of cell cluster-guided mutual nearest neighbor searching in improving the accuracy of batch effect correction. In addition, it is either a novel thing that a method can refine cluster labels and do batch effect correction simultaneously, for example, CarDEC by Lakkis et al. (2021). All these new methods have shown advantages in removing batch effect than the traditional methods, such as Seurat3, Harmony and scVI. Thus, authors should pay more attention to these new methods and more evaluations are required that comparing to Seurat3 and Harmony are not enough.

scDML calculates similarity of clusters based on KNN pairs within batch and MNN pairs across batches. However, when the batch effect between datasets gets large, that is the batch effect is not orthogonal to biological signals, the identified MNN pairs are not that reliable because some pairs may consist of cells from different cell types. However, it seems this impact on the accuracy of similarity computation is not considered by authors.

Other comments:

1. Since there are some new methods recently developed with similar idea to scDML, it is worthy to include them in the introduction section.
2. Authors used the number of true cell type as the cut-off for all datasets analyzed, and one evaluation was implemented to demonstrate the little impact of different cluster number selection

on the performance of scDML. However, it is not enough for such an important parameter. For most datasets, users have no idea on the true cluster numbers. More evidence is required to show the robustness of cluster number selection.

3. At line 107, full names are required when ARI, NMI etc first appear.

4. At line 199, authors mentioned that scDML can detect "novel cell subtypes", however, from their findings, it is hard to say they are novel ones, while they are only the subtypes missed by Seurat3, Harmony and scVI.

5. At line 238, authors showed the advantages of scDML in down-sampled data. However, it seems to there is no need to do such an analysis, because in the downsampled data, there is fewer cell types, especially fewer rare ones, this makes the case even easier than the original data. Authors should explain more to the logic of this analysis and why they did this analysis.

6. At line 267, authors made the conclusion that scDML could preserve the hierarchical structure of original data based on the results shown in Figure 6. However, I did not get the underlying logic. Are the hierarchical structure obtained from the output of scDML or from the true labels? If from scDML results, authors seems to demonstrate their advantages based on their own standard, and there is no evidence directly supporting the true structure is kept after analysis.

7. Citations are required for ARI, NMI, average silhouette width, iLISI and KL.

We thank the editor and reviewers for their careful reading and constructive comments. Below are our point-by-point responses to the reviewers' comments. The original reviewers' comments are in italics and our responses are in normal font colored in blue. The changes that we make in the main manuscript are tracked and highlighted in red.

REVIEWER COMMENTS

Reviewer #1 (Remarks to the Author):

Yu et al. presented a method named scDML for batch integration of single-cell transcriptomics data. The method is guided by the initial clusters and the nearest neighbors within and between batches. Using simulation and experimentally acquired datasets, the authors demonstrated that scDML outperformed competing methods.

We thank the reviewer for the comments and suggestions concerning our manuscript, which are all valuable and very helpful for revising and improving our paper, as well as the important guiding significance to our research. We have taken all these comments into account and made major corrections in the revised manuscript. The main changes are highlighted in red color for you to access easily.

Major concerns:

1. I am concerned that methods such as BBKNN, Harmony, Seurat3, scVI were not run properly or not run in their optimal ways. These methods are very popular and widely adopted by researchers. Several benchmark studies also demonstrated their superior performance on a wide range of applications. However, in Figure 2, Figure 4, Figure 5, the authors showed that these four methods performed poorly. Much better performance is expected from these methods.

Thank you for your concerns. We carefully check the script we used and followed respective official tutorials of these methods.

Firstly, we are sorry for that we did not use the updated tutorial for scVI but the tutorial in the previous version, in which we selected highly variable genes after `sc.pp.normalize_total` and `sc.pp.log1p` and fed the normalized counts into scVI. However, in the updated tutorial, scVI selects highly variable genes using raw counts directly and feeds the raw counts into scVI. Therefore, in our revised manuscript, we re-run scVI on all datasets based on its updated tutorial (https://docs.scvi-tools.org/en/stable/tutorials/notebooks/api_overview.html). Nevertheless, there was no difference in the performance of scVI before and after the change.

Secondly, we think the most controversial is harmony, because the results of the simulated dataset and the results of scib's benchmark

(https://theislab.github.io/scib-reproducibility/dataset_simulations_2.html) are totally different. So, we looked for the reason carefully, and the reason is that **scib** uses a different way of choosing highly variable genes and correction variables for harmony. We put the jupyter notebook for harmony into google colab (<https://colab.research.google.com/drive/1MQt8Cex65NjnO7fUH5GFrkjN8wtZnSx9?usp=sharing>), and allow you to interactively reproduce the result. It can be seen that harmony is hardly affected by the number of PCs and the number of highly variable genes (**Table R1**). For this simulation dataset, harmony only works well when correcting BATCH and SubBatch simultaneously, or choosing highly variable genes group_by BATCH and correcting SubBatch. However, for all other competing methods including scDML, we always choose HVGs across all datasets and correct the batch variable BATCH. Obviously, this is not fair to other methods if we choose the optimal case for harmony. So, we still use the same preprocessing procedure for selecting highly variable genes as other methods. Moreover, the real data analysis (Figure 3-Figure 7) shows that harmony has a relatively good performance, which in turn indicates that we used it properly.

On the other hand, we also tested scDML based on HVGs obtained by *scib* pipeline (group_by Batch, 1000 and 2000 HVGs used). The results show that scDML can exactly recover the number of true clusters (**Figure R1**), which demonstrates that scDML is robust to the way of choosing HVGs.

Table R1. Running Harmony with different parameters. The first column indicates whether using the preprocessing procedure in scib package. The second column indicates the number of highly variable genes we used. The third column means the number of PCs used. The fourth column means the way of choosing highly variable genes, for example, group_by=BATCH means we used `sc.pp.highly_variable_genes(adata, batch_key="BATCH")`. The fifth column means which variables to be corrected, for example, "BATCH, SubBatch" means that harmony corrects two variables simultaneously. The last column indicates the performance, in which "Bad" means harmony fails to correct batch effect and gets wrong clusters, and "Good" means harmony is able to remove batch effect and obtains the correct clusters.

	Number of HVGs	nPCs	HVGs group_by	Correct variables	Performance (visually)
Preprocessing by scib	1000	20	BATCH	BATCH	Bad
	1000	20	BATCH	SubBatch	Good
	1000	20	SubBatch	BATCH	Bad
	1000	20	SubBatch	SubBatch	Bad
	1000	50	BATCH	BATCH	Bad
	1000	50	BATCH	SubBatch	Good
	1000	50	SubBatch	BATCH	Bad
	1000	50	SubBatch	SubBatch	Bad
	2000	20	BATCH	BATCH	Bad

		2000	20	BATCH	SubBatch	Good
		2000	20	SubBatch	BATCH	Bad
		2000	20	SubBatch	SubBatch	Bad
		2000	50	BATCH	BATCH	Bad
		2000	50	BATCH	SubBatch	Good
		2000	50	SubBatch	BATCH	Bad
		2000	50	SubBatch	SubBatch	Bad
Preprocessing by normal pipeline		1000	50	None*	BATCH	Bad
		1000	50	None*	SubBatch	Bad
		1000	50	None*	BATCH, SubBatch	Good
		1000	50	BATCH	BATCH	Bad
		1000	50	BATCH	SubBatch	Bad
		1000	50	BATCH	BATCH, SubBatch	Good
		1000	50	SubBatch	BATCH	Bad
		1000	50	SubBatch	SubBatch	Bad
		1000	50	SubBatch	BATCH, SubBatch	Bad

*None means we used the function `sc.pp.highly_variable_genes` with `batch_key=None`.

Figure R1. The performance of scDML based on HVGS from scib pipeline. (A) 1000 HVGS group_by BATCH. (B) 2000 HVGS group_by BATCH.

Thirdly, we used the standard pipeline for Seurat (https://satijalab.org/seurat/articles/integration_introduction.html) and for BBKNN (<https://nbviewer.org/github/Teichlab/bbknn/blob/master/examples/simulation.ipynb>).

Lastly, for real datasets (for example, Figure 4), we do not think the competing methods are not run properly. Although the batch mixing index of harmony (BatchKL=0.809) and Seurat (BatchKL=0.784) is worse than that of scDML

(BatchKL=0.591), the ARI of Seurat (0.643) ranked the first, closely followed by scDML (0.631) and harmony (0.629). As for Figure 5 (data was downloaded from Fig 16 in Tran et al. PMID: 31948481), we compared the results in our paper with that in Fig 16 in Tran et al (2020). visually and found little difference between the two results.

Overall, we updated the results of BBKNN from previous 1000 HVGs to 2000 HVGs. And for scVI, we changed the way of selecting highly variable genes (according to scVI’s official tutorial, https://docs.scvi-tools.org/en/stable/tutorials/notebooks/api_overview.html) using `sc.pp.highly_variable_genes` (`adata`, `batch_key="BATCH"`, `flavor="seurat_v3"`, `layer="counts"`), and fed the raw counts as the input of scVI. Also, we added an extra competing method CarDEC (PMID: 34035047) as the comparison.

All figures and text have already been adjusted accordingly. Please refer to our revised manuscript for details.

2. Is scDML able to integrate large numbers of batches/samples? For instance, constructing a covid atlas from more than 1000 samples/patients. What is the upper limit of number of batches/sample?

Thank you for this helpful question. For large number of batches/samples, the time of finding MNN pairs will increase quadratically, but in our revised scDML algorithm we accelerated it by parallel computing. Moreover, to evaluate the performance of scDML on large samples/batches, we firstly added a healthy human heart atlas, which has 485193 cells from 140 batches with strong batch effect. scDML finished the job in about 160 minutes and obtained the highest ARI (0.763), followed by harmony, and meanwhile removed the batch effect (Figure R2).

Figure R2. scDML enables integrating large number of batches and simultaneously removing batch effect for the human heart dataset with 140 batches. The left panel: bar plot shows the value of ASW_celltype and ASW_batch, in which the bar height denotes the value of ASW_celltype and the point height denotes the value of ASW_batch. Higher ASW_celltype and lower ASW_batch means better performance. The middle panel: scatter plot shows the value of BatchKL (x-axis) and iLISI (y-axis). Point closer to the upper right means better performance. The right panel: bar plot shows the value of ARI and NMI for different methods. Higher bar means better performance.

To demonstrate that scDML is able to remove multi-level batch effect, we also included a failing heart atlas from two sequencing technologies (single cell and single nuclei RNA-seq) and in total 45 samples. scDML reached the highest ARI (0.975), followed by harmony (0.930), and can remove batch effect from different technologies and samples (Figure R3, and Figure S13 in revised manuscript).

Figure R3. The results of scDML on failing heart atlas from two sequencing technologies (single cell and single nuclei RNA-seq) and in total 45 samples.

As we discussed in Question 2 of Reviewer 3 (Table R2), scDML has a relatively high tolerance rate for false MNN pairs. Therefore, scDML is able to integrate large numbers of samples.

3. In most of the figures, UMAP plots before batch integration should be added to show the presence of batch effects.

Thank you for the helpful suggestion. We have added the UMAP plots without batch integration. Please see our revised manuscript and figures.

4. In Figure 3, UMAP plots should be shown the main figure, just like Figure 2, 4, 5.

Thank you for the helpful suggestion. We have adjusted and rearranged the figures accordingly.

5. The Sankey plots in Figure 3A are misleading. The authors stated that there are much more crossings in the Sankey plots of competing methods (Line 172). These crossings can be easily removed by reordering the clusters. For instance, in the harmony plot, if we move cluster 4 to between 1 and 3, the Sankey plot will have less crossings.

Thanks for raising this important point. The sankey plots were automatically output by R package *googleVis* and there is indeed misleading issue due to the order of clusters, so we removed the statements “more crossings”. Although ARI and NMI can measure the performance of different integration methods, Sankey plots can more intuitively show the consistency between identified clusters and true cell types, especially for some tiny cell types. For example, in harmony’s Sankey plot, even if we

move cluster 4 between 1 and 3, we still cannot distinguish some tiny clusters, such as *activated_stellate* and *quiescent_stellate*. In our revised manuscript, we only use Sankey plots to highlight the consistency between clusters and true cell types.

Minor comments:

1. Line 98, reference is required for BERMUDA.

Thanks for raising this important point. We have added it in our revised manuscript.

We hope that you are satisfied with our revisions. Thanks again for all your constructive suggestions.

Reviewer #2 (Remarks to the Author):

In "Batch alignment of single-cell transcriptomics data using deep metric learning," Yu et al. propose a new method for scRNA-seq integration using a multistep pipeline involving clustering, MNN search to find anchors, and deep metric learning using a small neural network with triplet loss to find an integrated embedding. Overall, I found the method sound and interesting in principle, with a good amount of benchmarking using appropriate metrics, and that shows the method performs comparably well against existing tools. I therefore recommend publication of the manuscript after minor revisions, which are noted below

- What are the noteworthy results?

The process of high-resolution clustering, finding MNNs based on these clusters (and potentially merging them), followed by triplet loss is a good approach. The benchmarking results look promising.

- Will the work be of significance to the field and related fields? How does it compare to the established literature? If the work is not original, please provide relevant references.

The method could be another useful one to try among the current suite of integration tools. The authors provide good comparison to existing literature. The method makes use of existing approaches, e.g., Haghverdi et al. (2018) introduced MNNs, clustering algorithms like Louvain are common in scRNA-seq, triplet loss and metric learning has been used in a number of studies like Simon et al., 2021, though the particular pipeline and implementation are original.

- Does the work support the conclusions and claims, or is additional evidence needed?
Yes

- Are there any flaws in the data analysis, interpretation and conclusions? - Do these prohibit publication or require revision?

No

- *Is the methodology sound? Does the work meet the expected standards in your field?*
Yes

- *Is there enough detail provided in the methods for the work to be reproduced?*
Authors do mostly a good job, a few points noted below.

We thank the reviewer for the positive comments about our manuscript. We have taken all these comments into account, and have made corrections in the revised manuscript. The main changes are highlighted in red color for you to access easily.

Other comments

1. *Page 1, line 20: “unappreciated levels of heterogeneity”: This is a vague statement and the sentence could just be deleted.*

Thanks for raising this important point. We have deleted it in our revised manuscript.

2. *Line 21: “accurately detecting the number of cell types”: This problem is not rigorously addressed by the authors and so this phrase should be considered for removal.*

Thank you for the helpful suggestion. We have removed this phrase in our revised manuscript. What we want to express here is that the number of clusters identified by scDML is most consistent with the number of real cell types.

3. *Page 3, lines 56-7: My understanding of the benchmarking study of Luecken et al. (reference 12) is that the neural architectures (scVI, SCANVI) perform best overall, especially for major dataset differences, with Scanorama performing better for subtle differences. I am also not sure what “over-denoised” means in line 64. The authors should comment on this.*

Thanks for your the helpful suggestions. In the benchmark study by Luecken et al, the performance of harmony almost ranked the first for small datasets, for example, the pancreas and simulation dataset, while scANVI and scVI performed the best for larger data, for example the lung atlas and mouse brain data (<https://theislab.github.io/scib-reproducibility/>). But in the other two benchmark studies (PMID: 31948481 and PMID: 33524142), they recommended Harmony as the first method to try. Based on our limited experiences, although Scanorama performs better for some situations, it cannot mix different batches very well. Moreover, the semi-supervised mode of scANVI and extremely time-consuming scVI hinder the application in real datasets. We rewrote the introduction section in our revised manuscript, and conducted a more comprehensive literature review.

As for “over-denoised” in line 64, we mean that in the denoised counts from scVI or CarDEC (output by the decoder layer), almost all zero values turn into non-zero, and if we use the denoised counts to do differential expression analysis, it will cause serious false positive rate (Xu et al, PMID: 35821114).

4. Line 72: “make the clusters as many as possible”: The authors should just say something like “clustered at a high resolution (Louvain resolution of 3.0)” or something to that effect. As is, this is not an accurate statement (i.e., maximizing the clusters would theoretically just put each cell into its own cluster).

Thanks for your constructive suggestion. At the initial step, scDML needs to cluster cells at a high resolution and then used the presented merge rule to merge these clusters. In our revised manuscript, we removed some inaccurate expressions and modified the statements accordingly. We hope that you are satisfied with our revisions.

5. Page 4, line 78: “serval” should be “several

Thanks for pointing this out. We have proofread the paper and corrected such typos.

Page 11, line 248: “significantly” should be “substantially” to avoid confusion with statistical significance

Thanks for reminding us of this important point. We have revised the paper accordingly.

7. Page 22, lines 479-486 and Algorithm 2: I was unable to find important details on the architecture of the embedding network. Aside from the loss and the numbers of nodes per layer, which are provided, the authors should also provide the activation functions, connectivity structure, etc.

Thank you for the helpful suggestion. We have added description of the embedding network architecture, activation function, loss function, connectivity structure and the number of nodes per layer in Supplementary Notes before Algorithm 2.

We hope that you are satisfied with our revisions. Thanks again for all your constructive suggestions.

Reviewer #3 (Remarks to the Author):

In this manuscript, Yu et al. presented a deep metric learning-based method scDML for correcting the batch effect among single-cell RNA-seq data. scDML borrows the initial cluster information to refine the clustering and remove batch effect. According

to the evaluation results, scDML outperformed the state-of-the-arts methods, such as Seurat, Harmony, scVI etc. However, there are still some concerns that have to be addressed before any further consideration.

We thank the reviewer for the comments and suggestions concerning our manuscript, which are all valuable and very helpful for revising and improving our paper, as well as the important guiding significance to our research. We have taken all these comments into account, and have made major corrections in the revised manuscript. The main changes are highlighted in red color for you to access easily.

Major comments:

Now methods performing batch effect with the help of cell cluster information are now something really new. For example, SMNN and iSMNN by Yang et al. (2021; 2021) both take the advantage of cell cluster-guided mutual nearest neighbor searching in improving the accuracy of batch effect correction. In addition, it is either a novel thing that a method can refine cluster labels and do batch effect correction simultaneously, for example, CarDEC by Lakkis et al. (2021). All these new methods have shown advantages in removing batch effect than the traditional methods, such as Seurat3, Harmony and scVI. Thus, authors should pay more attention to these new methods and more evaluations are required that comparing to Seurat3 and Harmony are not enough.

Thank you for your instructive suggestions. Although both SMNN and iSMNN take the advantage of cell cluster-guided mutual nearest neighbor searching in improving the accuracy of batch effect correction, they are actually a supervised method. Since SMNN is only a special case of iSMNN, we only tested iSMNN. Even if we set the true cell type label as the guided information, the performance of iSMNN on bct dataset is still not satisfactory (Figure R4), as cells from different batches did not mix well. For bct_del dataset (cell type *basal* only in batch *spk*), iSMNN only used the two common cell types in three batches, so the result is even worse (Figure R5). For simulation dataset (data analyzed in Figure 2), the result is still unsatisfactory. Therefore, in our revised manuscript, we only added CarDEC as the comparison.

We have updated all figures in the revised manuscript to show the new comparison results. Please refer to the Results section for more details.

Figure R4. UMAP shows the results of iSMNN using the true cell type as the guide information on *bct* dataset. The left is colored by batch, and the right is colored by cell type.

Figure R5. UMAP shows the results of iSMNN using the true cell type as the guide information on *bct* dataset with *basal* removed from batch *vis* and *wal*. The left is colored by batch, and the right is colored by cell type.

Figure R6. UMAP shows the results of iSMNN using the true cell type as the guide

information on simulation dataset (data analyzed in Figure 2 in our manuscript). The left is colored by batch, and the right is colored by cell type.

scDML calculates similarity of clusters based on KNN pairs within batch and MNN pairs across batches. However, when the batch effect between datasets gets large, that is the batch effect is not orthogonal to biological signals, the identified MNN pairs are not that reliable because some pairs may consist of cells from different cell types. However, it seems this impact on the accuracy of similarity computation is not considered by authors.

Thank you for your instructive suggestion. To demonstrate the robustness and effectiveness of scDML, we calculate the accuracy of MNN pairs given the true cell type label, which is defined as

$$\text{ACC}(\text{total MNNs}) = \frac{n}{N} * 100\%,$$

where N is the total number of MNN pairs, and n is the number of MNN pairs (i,j) that cell i and cell j belong to the same cell type. The following table reports the accuracy of total MNN pairs for all datasets we analyzed.

Table R2. The accuracy of total MNN pairs for all datasets we used. The accuracy is computed based on the true cell type label. Larger value means more accurate MNN pairs found.

Dataset	ACC(total MNNs)
bct	0.9959
bct_del	0.9737
simulation1	0.9818
simulation2	0.9563
pancreas	0.9735
Macaque retina	0.9413
Mouse retina	0.9901
FullMouseBrain	0.9606
Healthy heart atlas (140 batches)	0.6765
Failing heart dataset (multi-level batch, 45 batches)	0.9632
Lung_two_species	0.5500

Although the accuracy of total MNN pairs for lung_two species is relatively low, the final performance of scDML is still the best. For other datasets, the relatively high total accuracy of MNN pairs guarantees that the final merged clusters are correct. This in turn shows that scDML has a relatively high tolerance rate for false MNN pairs.

Since we cannot measure the batch effect of real data in advance, we use the R package splatter (PMID: 28899397) to simulate data (4000 cells and 10000 genes

from two batches, in which each batch has 2000 cells, and fix parameter $group.prob=c(0.25,0.25,0.25,0.25)$) to prove that scDML is robust to MNN pairs when batch effect gets larger. Specifically, we set different values for the parameter $batch.facLoc$ and $batch.facScale$ to simulate datasets with different batch effect level (Table R3).

Table R3. The accuracy of total MNN pairs for dataset simulated by splatter with different batch parameters.

Parameter	ACC (total MNNs)	ARI	NMI
batch.facLoc=4.0, batch.facScale=4.0	0.3006	0.25	0.47
batch.facLoc=3.0, batch.facScale=3.0	0.3756	0.9637	0.9377
batch.facLoc=2.0, batch.facScale=2.0	0.5678	0.9967	0.9931
batch.facLoc=1.0, batch.facScale=1.0	0.9318	1.0	1.0

The reason why the ARI and NMI is low when $batch.facLoc=4.0$, $batch.facScale=4.0$ is that the batch effect is much larger than the biological difference. The distance between cell types is almost impossible to distinguish (Figure R7). As the batch effect is gradually reduced, even if the accuracy of total MNN pairs is not very high, the ARI and NMI are very close to 1.0. This further demonstrates the robustness of scDML.

Figure R7. Dataset simulated by splatter with parameters $batch.facLoc=4.0$ and $batch.facScale=4.0$. The left panel is colored by batch and the right panel is colored by the true cell type. We can see that this dataset has very strong batch effect and true cell types mix together. All popular batch corrected methods, such as Seurat and harmony, also fail to correct its batch effect.

Other comments:

1. Since there are some new methods recently developed with similar idea to scDML, it is worthy to include them in the introduction section.

Thank you for your instructive suggestion. We rewrote the introduction section and included some new methods focused on clustering and batch effect removal, such as SMNN, iSMNN, CarDEC et al.

2. Authors used the number of true cell type as the cut-off for all datasets analyzed, and one evaluation was implemented to demonstrate the little impact of different cluster number selection on the performance of scDML. However, it is not enough for such an important parameter. For most datasets, users have no idea on the true cluster numbers. More evidence is required to show the robustness of cluster number selection.

Thank you for your insightful comment. In practice, we have no idea on the true number of clusters. For popular clustering algorithm, for example, Louvain and Leiden, we can set different resolution to obtain different number of clusters and manually merge them based on biological knowledge. scDML primarily constructs a similarity matrix and presents a hierarchical merge rule for initial clusters. Inspired by Zelnik-manor et al (<https://dl.acm.org/doi/10.5555/2976040.2976241>), and the idea of spectral clustering, we provided a strategy to automatically infer the number of clusters with eigenvalues of the similarity matrix or manually set the number of clusters based on the heat map of similarity matrix.

Figure R8. The similarity matrix for bct (A), bct_del (B), macaque(C).

For example, we can easily set the number of optimal clusters to be 3 for simple dataset we analyzed (Figure R8A). For complex datasets (Figure R8B, Figure R8C), we can use the theory of the eigenvalue of similarity matrix to give the appropriate number of clusters. For bct_del dataset, the spectral clustering algorithm suggests the optimal number of clusters is [2,3,6,14]. For macaque dataset, the spectral clustering algorithm suggests the optimal number of clusters is [8,12,18,5]. Although the first suggested optimal number of clusters is 8, the second suggested optimal number of clusters is 12, which exactly equals to the number of true cell types. Nevertheless, in the real data analysis, we still need biological knowledge, and marker genes to determine how many classes are finally defined.

Additionally, we also evaluate the performance of scDML on macaque retina data

with varying parameters, including three different resolutions for Louvain algorithm (3.0, 6.0, 9.0), two different numbers of highly variable genes (1000 and 2000), three numbers of clusters after merging (11, 12 and 13), and 21 different combinations for (K_bw, K_in). We can see that scDML is very robust to the choice of hyper-parameters (Figure R9).

Figure R9. ARI (the left panel) and NMI (the right panel) of scDML with three different resolutions for Louvain algorithm (3.0, 6.0, 9.0), two different numbers of highly variable genes (1000 and 2000), three numbers of clusters after merging (11, 12 and 13), and 21 different combinations for (K_bw, K_in). Each bar represents a combination of (K_bw, K_in). There are 18 cases with respect to each bar, where the bar height is the mean value and the error bar is the standard deviation.

3. At line 107, full names are required when ARI, NMI etc first appear.

Thank you for the valuable suggestion. We amended them in our revised manuscript and provided necessary citations for all metrics we used.

4. At line 199, authors mentioned that scDML can detect "novel cell subtypes", however, from their findings, it is hard to say they are novel ones, while they are only the subtypes missed by Seurat3, Harmony and scVI.

We thank the reviewer for raising this point. Here we wanted to say that scDML can find some clusters missed by other methods. In our revised manuscript, we have restated that scDML can detect some tiny subtypes possibly missed by competing methods, such as Seurat3, Harmony and scVI.

5. At line 238, authors showed the advantages of scDML in down-sampled data. However, it seems to there is no need to do such an analysis, because in the downsampled data, there is fewer cell types, especially fewer rare ones, this makes the case even easier than the original data. Authors should explain more to the logic of this analysis and why they did this analysis.

Thank you for the valuable comment. At the very beginning, we wanted to prove the

stability of the proposed method using the full and down-sampling dataset. Although scDML was significantly improved with more adequate batch mixing and cleaner clusters in the down-sampling dataset compared to the full dataset, other competing methods hardly change for cell types such as Neuron. In our revised manuscript, we removed this down-sampled data and instead added two extra-large scRNA-seq datasets to show the superiority of our method.

6. At line 267, authors made the conclusion that scDML could preserve the hierarchical structure of original data based on the results shown in Figure 6. However, I did not get the underlying logic. Are the hierarchical structure obtained from the output of scDML or from the true labels? If from scDML results, authors seems to demonstrate their advantages based on their own standard, and there is no evidence directly supporting the true structure is kept after analysis.

Thank you for your inspiring question. Actually, scDML primarily presents a hierarchical merge rule for initial clusters based on Louvain or Leiden algorithm, using MNN and KNN pairs. The mouse retina dataset (PMID: 27565351) used in Figure 6 has a gold-standard hierarchical structure (Figure R10) from its original literature (PMID: 27565351), in other word, the hierarchical structure obtained from the true labels. Here we wanted to say that the hierarchical structure obtained from the output of scDML (Figure 6b) is consistent with that from the true labels (Figure R10), which demonstrates the advantage of the proposed merge rule. We have provided clearer illustration in the revised manuscript.

Figure R10. The hierarchical tree structure of mouse retina from its original literature (Shekhar et al 2016, PMID: 27565351).

7. Citations are required for ARI, NMI, average silhouette width, iLISI and KL.

Thank you for your instructive suggestion. We have provided necessary citations for ARI, NMI, ASW, iLISI and KL accordingly in our revised manuscript.

We hope that you are satisfied with our revisions. Thanks again for all your

constructive suggestions.

Reviewer #1 (Remarks to the Author):

Thanks the authors for the responses. I have no further questions or comments.

Reviewer #3 (Remarks to the Author):

Authors have fully addressed my concerns and I don't have any more questions on this manuscript.